# When is a handaxe a planned-axe? Exploring morphological variability in the Acheulean

**James Clark** [1]*, **Ceri Shipton**[2,3], **Marie-Hélène Moncel** [4], **Philip Ronald Nigst** [5,6], **Robert Andrew Foley**[1]

1 Department of Archaeology, Leverhulme Centre for Human Evolutionary Studies, University of Cambridge, Cambridge, United Kingdom, 2 Institute of Archaeology, University College London, London, United Kingdom, 3 Department of Archaeology, McDonald Institure for Archaeological Research, University of Cambridge, Cambridge, United Kingdom, 4 Département Homme et Environnement, UMR 7194 HNHP (MNHN-CNRS-UPVD), Muséum National d'Histoire Naturelle, Paris, France, 5 Department of Prehistoric and Historical Archaeology, University of Vienna, Vienna, Austria, 6 Human Evolution and Archaeological Sciences (HEAS), University of Vienna, Vienna, Austria

* jc2012@cam.ac.uk

## Abstract

The handaxe is an iconic stone tool form used to define and symbolise both the Acheulean and the wider Palaeolithic. There has long been debate around the extent of its morphological variability between sites, and the role that extrinsic factors (especially raw material, blank type, and the extent of resharpening) have played in driving this variability, but there has been a lack of high-resolution examinations of these factors in the same study. In this paper, we present a 2D geometric morphometric analysis of 1097 handaxes from across Africa, the Levant, and western Europe to examine the patterning of this variability and what it can tell us about hominin behaviour. We replicate the findings of previous studies, that handaxe shape varies significantly between sites and entire continental regions, but we find no evidence for raw material, blank type, or resharpening in determining this pattern. What we do find, however, is that markers of reduction trajectory vary substantially between sites, suggesting that handaxes were deployed differently according to hominin need at a given site. We argue this is reflective of a continuum of reduction strategies, from those focused on the maintenance of a sharp cutting edge (i.e. direct use in cutting activities), to those focused on maintaining tip shapes, and perhaps a corresponding production of flakes. Implications for hominin behavioural flexibility are discussed.

**Data Availability Statement:** All relevant data are within the manuscript and its Supporting Information files.

## 1. Introduction

The Acheulean is historically defined by the presence of large (commonly more than 10cm long), bilaterally symmetrical artefacts, usually worked on two sides (e.g. [1,2]), which first appear in the archaeological record after ~1.8 Million years ago (Ma) [3–6]. These Mode 2 technologies, as they are otherwise known [7], can be subdivided into (at least): cleavers (defined by a straight transverse cutting edge at the distal end), picks (with a thick trihedral or quadrangular distal tip), massive scrapers (steep unifacial retouch), and handaxes (e.g. [1]).

**Funding:** JC: Cambridge Trust UK Master's Scholarship (Grant number unknown), https://www.cambridgetrust.org/. Funder had NO role in the study design, data collection and analysis, decision to publish, or preparation of the manuscript. JC: University of Cambridge Department of Archaeology Studentship (Grant number unknown), https://www.arch.cam.ac.uk/. Funder had NO role in the study design, data collection and analysis, decision to publish, or preparation of the manuscript. JC: St John's College Graduate Scholarship (Grant number unknown), https://www.joh.cam.ac.uk/. Funder had NO role in the study design, data collection and analysis, decision to publish, or preparation of the manuscript.

**Competing interests:** The authors have declared that no competing interests exist.

We define handaxes as specimens with convergence of lateral edges to a tip (cf. [1]), but with a single plane of intersection between the two faces either side of the tip (cf. [8]). This form was originally designated as the *fossile directeur* of the Acheulean [9], and remains the Mode 2 morphotype found most consistently across time and space. Nonetheless, it is worth keeping in mind that Mode 2 technologies are not a consistent component of Acheulean-era assemblages, perhaps related to differences in cultural/biological history, site function, mobility, and/or ecology [10–17].

Handaxes are interesting precisely because of their long temporal and geographic persistence, and their implications for biological and cultural evolution in the Lower Palaeolithic. For example, the wide range of these tools may indicate an ability to survive in a broad range of environmental conditions, with shifts in habitat and climate necessary aspects of dispersal from equatorial to more temperate latitudes. Some authors have emphasised their shared characteristics in support of a biological influence in their manufacture (e.g. [18–20]). Others suggest that the substantial variability existing within handaxe manufacture indicates cultural transmission, perhaps alongside some adaptive cultural conservatism (e.g. [21–30]). A cultural mechanism for handaxe manufacture raises a number of questions, including the nature of their diffusion across time and space, how they aid adaptation to new and/or changing conditions, and their potential for reinvention [24,26,31,32].

Where variability does exist, authors frequently disagree on the extent to which this reflects an intentional imposition by the knapper. Amongst the possible extrinsic drivers of inter-assemblage variability, particular discussion has been given to raw material quality, blank type, and the extent of resharpening. With regards to raw material, White [33] has suggested that the use of small river cobbles and/or lower quality flint at British sites may have biased handaxe shapes towards more pointed forms, whereas larger and higher quality nodules from primary outcrops might have allowed for more intensive reduction and production of preferred ovate forms. In contrast, Eren et al. [34] found that replication of a target handaxe form on three different raw materials (flint, basalt, and obsidian) resulted in no significant differences in final form, suggesting limited influence of raw material on handaxe morphology. While also necessarily related to raw material types, Sharon [35–37] has put greater emphasis on the type of blank used in biface manufacture, specifically through the existence of the "Large Flake Acheulean" in large swathes of Africa, southern Asia, and southern Europe. This is said to correspond to where coarse-grained rocks are widely available, in contrast to the more frequent use of fine-grained materials in northwestern Europe, which are less readily available in large boulders. This may have had a role in driving the morphological variability of handaxes, especially in the limited working of the ventral face when using a flake blank, and the increase in pointed forms [36,37].

McPherron [38–42] (see also [8]) has argued that a large proportion of the variation in handaxe shape both within and between sites (and regions) can be attributed to the influence of resharpening. This model suggests that handaxes become progressively rounded at the tip as the tip is reduced, reflecting the functional constraints of maintaining a useable cutting edge [8,38–40,42]. This conceptualises pointed assemblages as being only minimally resharpened and the most ovate assemblages as being the most resharpened. Shipton and Clarkson [43] have instead suggested that this relationship might be simply caused by rounded specimens requiring a greater initial reduction of the blank to achieve the target form, rather than a continuous rounding over the use-life of the specimen.

A growing body of authors are also emphasising that variability in handaxe morphology over time and space may reflect variation in norms of manufacture, and indeed the identities of their makers. Derek Roe [44,45] was an early proponent of using handaxe shapes in the British archaeological record as markers of cultural identity during the Acheulean, and these

patterns have been revived and revised as a strong chronological framework has been developed for these sites (e.g. [46–50]). Similarly, recent data on French, and Iberian assemblages have suggested that regional differences in biface manufacture can be seen between different regions of Europe in the Middle Pleistocene [51].

In this context, it is crucial to try and distinguish between the sources of handaxe variability at these different scales, ranging from intra-site drivers of morphological change, to broad scale changes over time and space. This may better allow us to understand the functional, demographic, and cultural evolutionary processes behind handaxe manufacture and transmission, and ultimately reasons for handaxe persistence over time [52]. However, this can only be achieved by inference of the survival strategies of the hominins that produced and used them. This study (from the graduate work of JC) attempts to directly address these questions with an analysis of a large (n = 1097) handaxe dataset, comprising specimens from across East Africa, the Levant, and Europe. These specimens are subject to a comparison of shape between regions, with an attempt to account for the impacts of raw material, blank type, and the extent of resharpening at individual sites. We also examine the wider trajectory of reduction at each site, in an attempt to understand the adaptive basis (or bases) of handaxe manufacture. We discuss the implications of our results for understanding the Acheulean more broadly.

## 2. Materials and methods

### 2.1. Sample

Data were collected from plan- and profile-view photos or drawings of a total of 1097 handaxes across 45 assemblages (26 site complexes) from Europe ($n_{total}$ = 478), the Levant ($n_{total}$ = 355), and eastern Africa ($n_{total}$ = 264). Samples were targeted to get a representative sample of variation in each continental region, but ultimately reflect those that could be accessed and analysed within the time frame of the project. This inevitably leaves spatial (e.g. for Iberia) and chronological (e.g. for the African Middle Pleistocene) gaps in the coverage of assemblages, but nonetheless resulted in a robust sampling of the three main area. The distribution of sites is shown in Fig 1, and each assemblage summarised in Table 1. These images were collected from a variety of sources, including the study of a number of museum collections, existing publications, and the work of collaborators. Following [25], samples from individual assemblages were capped at 30 so that within-sample variance was not distorted by sample size. Any intra-assemblage statistics were only calculated where individual sample sizes were ≥15. Where data were available, the raw material and blank type were recorded for each artefact. The midpoint of possible MIS stages was taken as the age-point of each site for preliminary temporal analyses. For more detailed descriptions of site stratigraphy, dating, sample composition, and technological data, see the Supplementary Information. Ethical approval for the study was granted by the University of Cambridge Department of Archaeology.

### 2.2. Measurement

It is crucial for morphometric (and especially geometric morphometric) analysis that handaxes are oriented according to a standardised procedure before measurements are taken so that measurements correspond to homologous points between specimens (e.g. [8,58–60]). As such, the images for each site were taken (where applicable) and processed according to a standardised protocol, described in the Supplementary Information.

A series of metrics were recorded for each artefact from the handaxe images, including length, width, thickness, and tip length (length from the tip to point of maximum width [44]), as well as the FlipTest index of asymmetry about both the length (longitudinal asymmetry) and width (latitudinal asymmetry) axes in plan view [61]. Discrepancies between computer

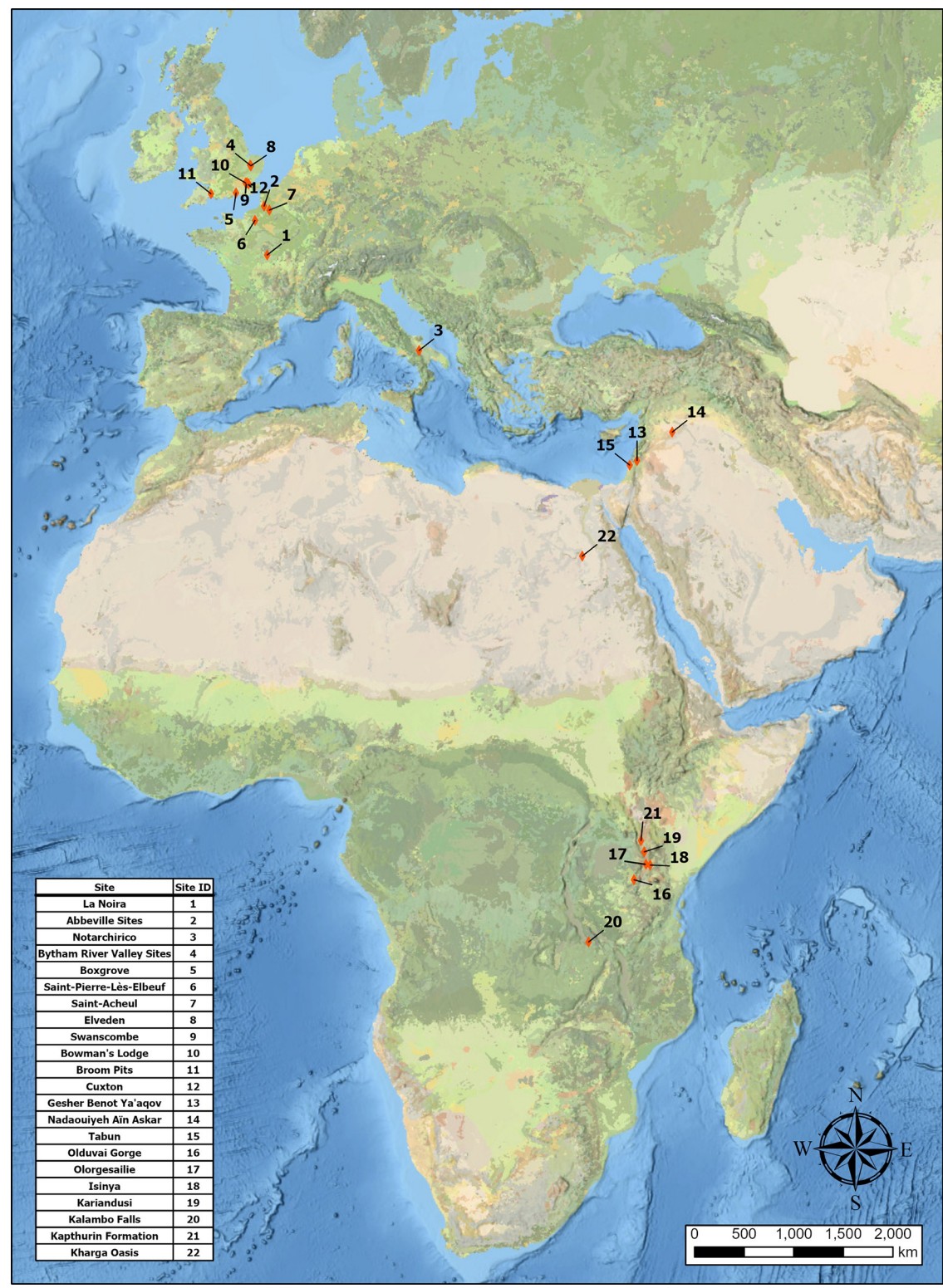

**Fig 1. Distribution of sites included in the investigation.** Map kindly produced by Jonathan Lim using ArcGIS Software by ESRI. The basemap was supported by ESRI under a license, original Copyright 2020 ESRI. Basemap source: National Geographic, Esri, DeLorme, HERE, UNEP-WCMC, USGS, NASA, ESA, METI, NRCAN, GEBCO, NOAA, iPC.

| Site | Site ID |
|------|---------|
| La Noira | 1 |
| Abbeville Sites | 2 |
| Notarchirico | 3 |
| Bytham River Valley Sites | 4 |
| Boxgrove | 5 |
| Saint-Pierre-Lès-Elbeuf | 6 |
| Saint-Acheul | 7 |
| Elveden | 8 |
| Swanscombe | 9 |
| Bowman's Lodge | 10 |
| Broom Pits | 11 |
| Cuxton | 12 |
| Gesher Benot Ya'aqov | 13 |
| Nadaouiyeh Aïn Askar | 14 |
| Tabun | 15 |
| Olduvai Gorge | 16 |
| Olorgesailie | 17 |
| Isinya | 18 |
| Kariandusi | 19 |
| Kalambo Falls | 20 |
| Kapthurin Formation | 21 |
| Kharga Oasis | 22 |

**Table 1. Summary of handaxes analysed in the present study.**

| Region | Assemblage | Country | Data Source(s) | MIS Stage(s) | Raw Material(s) | n |
|---|---|---|---|---|---|---|
| **Africa** | Olduvai Gorge EF-HR | Tanzania | 1 | MIS59-34 | Lava, Quartzite | 14 |
| | Olduvai Gorge TK | Tanzania | 1 | MIS41-34 | Quartzite, Lava | 8 |
| | Olduvai Gorge BK | Tanzania | 1 | MIS41-34 | Lava, Quartzite | 14 |
| | Olorgesailie Member 1 Site CL1-1 | Kenya | 2 | MIS36-25 | Basalt | 5 |
| | Olduvai Gorge Bed III (Assorted Sites) | Tanzania | 3 | MIS35-21 | Quartzite, Andesite, Nephelinite | 27 |
| | Olorgesailie Member 1 Site I3 | Kenya | 2 | MIS29-26 | Phonolite, Basalt | 15 |
| | Isinya | Kenya | 2 | ≥MIS24 | Phonolite, Consolidated Tuff, Chert, Quartz | 22 |
| | Olorgesailie Member 6/7 | Kenya | 2 | MIS27-23 | Phonolite | 11 |
| | Kariandusi | Kenya | 4 | MIS27-19 | Trachyte, Phonolite, Trachyte, Obsidian | 27 |
| | Olduvai Gorge HK | Tanzania | 3 | MIS28-18 | Quartzite, Andesite | 30 |
| | Olduvai Gorge MK | Tanzania | 3 | MIS28-18 | Quartzite, Basalt, Phonolite | 16 |
| | Olduvai Gorge MNK | Tanzania | 3 | MIS28-18 | Quartzite, Andesite | 18 |
| | Kalambo Falls Site B, Horizon V | Zambia | 5 | MIS13-10 | Quartzite, Silcrete, Chert | 24 |
| | Kapthurin Formation Site LHA/GnJh-03 | Kenya | 6 | MIS13-8 | Lava | 10 |
| | Kharga Oasis | Egypt | 4 | MIS11/9 | Chert | 23 |
| **Levant** | Gesher Benot Ya'aqov II-6 L4b | Israel | 7 | MIS18 | Basalt | 30 |
| | Gesher Benot Ya'aqov II-6 L4 | Israel | 7 | MIS18 | Basalt | 30 |
| | Gesher Benot Ya'aqov II-6 L1 | Israel | 7 | MIS18 | Basalt | 30 |
| | Nadaouiyeh Aïn Askar F | Syria | 4 | ≥MIS12 (MIS13?) | Flint | 30 |
| | Nadaouiyeh Aïn Askar E | Syria | 4 | ≥MIS12 (MIS13?) | Flint | 30 |
| | Nadaouiyeh Aïn Askar D | Syria | 4 | ≥MIS12 (MIS13?) | Flint | 30 |
| | Nadaouiyeh Aïn Askar C | Syria | 4 | ≥MIS12 (MIS12?) | Flint | 30 |
| | Tabun F | Israel | 3 | MIS13-11 | Flint, Chert | 25 |
| | Nadaouiyeh Aïn Askar B | Syria | 4 | ≤ MIS12 (MIS11?) | Flint | 30 |
| | Tabun Ed | Israel | 3 | MIS11 | Flint, Chert | 30 |
| | Tabun Eb | Israel | 3 | MIS10-8 | Flint, Chert | 30 |
| | Nadaouiyeh Aïn Askar A | Syria | 4 | ≤ MIS12 (MIS7?) | Flint | 30 |
| **Europe** | La Noira Lower | France | 4 | MIS16 | Millstone | 30 |
| | Moulin Quignon | France | 4 | MIS16 | Flint | 5 |
| | Notarchirico | Italy | 4 | MIS16 | Chert, Limestone, Quartzite | 20 |
| | Brandon Fields | UK | 4 | MIS15 | Flint | 30 |
| | Maidscross Hill | UK | 4 | MIS15 | Flint | 30 |
| | Carrière Carpentier | France | 4 | MIS14 | Flint | 5 |
| | Warren Hill Worn | UK | 3 | MIS13 | Flint | 30 |
| | Warren Hill Fresh | UK | 3 | MIS13 | Flint | 30 |
| | High Lodge | UK | 4 | MIS13 | Flint | 30 |
| | Boxgrove | UK | 3 | MIS13 | Flint | 30 |
| | La Noira Upper | France | 4 | MIS11 | Millstone, Chert, Flint | 30 |
| | Saint-Pierre-Lès-Elbeuf | France | 4 | MIS11(-5) | Flint | 30 |
| | Saint-Acheul | France | 4 | MIS11c | Flint | 30 |
| | Elveden | UK | 4 | MIS11c | Flint | 30 |
| | Swanscombe Middle Gravels | UK | 4 | MIS11c | Flint | 30 |
| | Bowman's Lodge | UK | 3 | MIS11a | Flint | 28 |
| | Broom Pits | UK | 3 | MIS9-8 | Chert, Flint | 30 |
| | Cuxton | UK | 3 | MIS9-8 | Flint | 30 |

*(Continued)*

**Table 1.** (Continued)

| Region | Assemblage | Country | Data Source(s) | MIS Stage(s) | Raw Material(s) | n |
|---|---|---|---|---|---|---|
| | | | | | **Total** | **1097** |

Key: 1 = [10], 2 = Data collected by CS from specimens in [53], 3 = [54], 4 = Data collected by JC for this Study, 5 = [55], 6 = [56], 7 = Data collected and shared by Gadi Herzlinger and Naama Goren-Inbar from specimens in [57]. There is an unfortunate dearth of assemblages on coarse-grained materials from Europe, especially from the Iberian peninsula, as well as of eastern African assemblages from the Middle Pleistocene.

and calliper measurements occurred between data sources. These were corrected for using a series of sample-specific OLS regression models to bring measurements in line with those made by callipers in order to maximise comparability between samples. The details of these models are described in the Supplementary Information. Final measurements were also used to generate ratios for both refinement (thickness/width) and elongation (width/length) [45].

## 2.3. Geometric morphometric analysis

Geometric Morphometric Analysis was carried out on all plan-view images, with the workflow of preparation for this is summarised in Fig 2. We are aware that not applying the analysis to profile images removes power from the analysis, especially when discriminating between tool forms [e.g. 60,62,63], as it does not account for the third dimension of volume. Nonetheless, we follow an established tradition of 2D biface shape analysis [e.g. 8,48,59,64–66]. Individual. tps files were created for the plan-view silhouettes from each site in tpsUtil-1.76, before a series of 60 equidistant outline coordinates (following [8,60])—starting from the tip—were saved for each artefact in tpsDig2-2.31 [67,68]. Raw coordinate data for each site were then combined in tpsUtil-1.76 and opened in PAST3 [69], where a 2D Procrustes Superimposition was carried out to further standardise the position, size, and orientation of each set of landmarks [70].

These transformed coordinates were then subjected to an Elliptical Fourier Analysis (EFA) in PAST3. EFA is a 2D geometric morphometric procedure that has been successfully applied to the study of handaxes [8,48,60,64]. EFA reduces closed curves into an infinite series of harmonics, each comprising four trigonometric Fourier coefficients [8,48]. The accuracy of the outline replication increases with each harmonic, but only the first N/2 harmonics should be

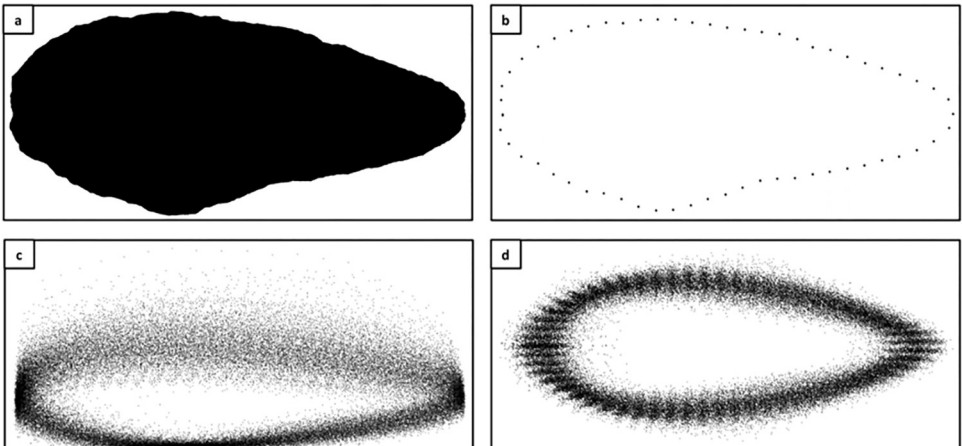

**Fig 2. Workflow in preparation for geometric morphometric analysis.** a: Handaxe silhouette, b: Single handaxe outline, c: Raw coordinate data, d: Procrustes-transformed landmark data.

used for further analysis, where N is the number of landmarks [69]. EFA presents a number of advantages over other 2D geometric morphometric techniques, with the most relevant here being the lack of a need for landmarks to be homologous between specimens [8,48]. EFA was carried out on the Procrustes-transformed coordinate data in PAST3, with the first 30 harmonics taken as the input for EFA Principal Component Analysis (PCA).

The principal component (PC) data was exported from PAST3, with statistical analyses carried out in SPSS 26 and RStudio 2022.12.0.353 running R 4.2.2 [71–73]. Graphs were created using the ggplot2 package for R [74]. All landmark, morphometric, and principal component data are available as supplementary information to this manuscript.

## 2.4. Allometry

Allometric relationships between shape and size were examined to explore changes to handaxe form over the course of knapping. In particular, McPherron [38–40,42] argues that handaxe resharpening results in a trajectory in which the tip is progressively rounded as it is reduced, reflecting the functional constraint of maintaining a functional cutting edge (see also [8]). Many authors have subsequently used PC scores against length tip length as a proxy for tracking the contribution of resharpening to an assemblage (e.g. [8,43,48,60,75]. McPherron [40] has also suggested this may explain previous findings of difference in handaxe morphology between regions.

In the present study, we explored the relationship between the PC scores and tip length at the level of the whole sample, as well as between regions and individual assemblages to see how such allometric relationships fit into the broader picture of inter-assemblage variability. We also used site-level allometric relationships to begin to explore the functional implications of changes to shape throughout knapping, that explicitly takes the production and maintenance (or not) of the cutting edge into account [52,53].

## 3. Results

The first 23 PCs from the EFA PCA account for >99% of shape variance, but PCs 1–3 are the only components that account for more than 5% (Fig 3). PC1 alone is responsible for 66.4%. PC1 represents elongation and tip shape, matching multiple previous investigations of handaxe shape (e.g. [48,59]). Higher scores reflect more elongated and pointed handaxes, and lower scores less elongated and more ovate forms. PC1 correlates very highly with metrical elongation from the whole sample (r = -.930, p ≈ 0.0E0), and Roe's [44,45] measure of tip shape (width at upper fifth over width at lower fifth) for the Nadaouiyeh handaxes [76] (r = -.533, p = 6.3314E-12). An OLS regression model indicates that, together, these two variables account for 94% of PC1 variance in the Nadaouiyeh subset.

PC2 (13.5% of variance) reflects lateral skew of raw material, with increasing magnitude of scores either side of 0 reflecting deviation from perfect (plan) symmetry. The absolute value of PC2 is significantly correlated with the longitudinal asymmetry index (r = .786, p = 3.7868E-231). That this variable only explains around 62% of the variation in asymmetry is likely a result the 60 coordinates simplifying the outline, and therefore removing sources of noise that may emerge between the individual points. Finally, PC3 (7.3% of variance) accounts for the position of maximum width on the handaxe, with higher scores suggesting maximum width close to the butt and lower scores placing it further towards the tip. This dimension correlates significantly with the ratio of tip length to total length (r = .663, p = 6.5606E-140). The remaining variability across the first 23 PCs is summarised in S1 Table in S1 File.

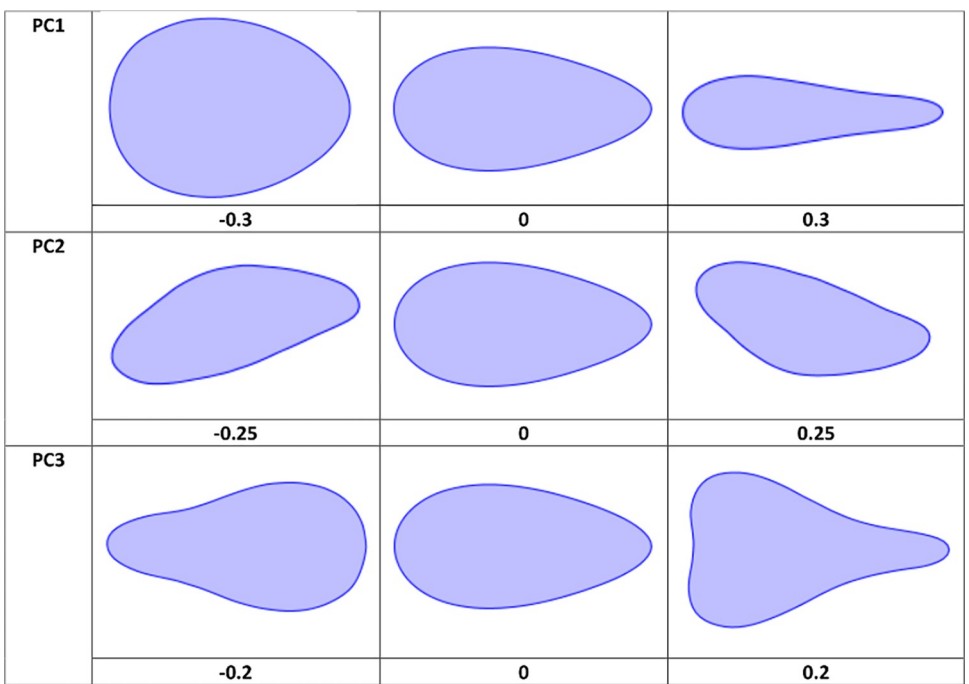

**Fig 3. Visualisation of shape variation represented by PCs 1–3 on the EFA PCA.** PC1 reflects elongation and tip shape, PC2 longitudinal asymmetry, and PC3 the position of maximum width.

## 3.1. Regional variation

**3.1.1. Plan shape.**   As PC2 is closely related to plan symmetry, which is presented separately, Fig 4 shows the shape distribution for each region on PC1 and PC3, as in [77]. This shows that, while there is clearly a substantial amount of overlap, the regional averages can clearly be separated on both dimensions, and especially PC1. African handaxes are the most pointed and elongated, Levantine handaxes the most ovate and least elongated, and European handaxes fall in the middle. Furthermore, Europe and especially the Levant show more positive extensions of variation within PC3 values, suggesting that the average position of maximum width lies closer to the butt than in African handaxes. One-way ANOVAs confirm that both PC1 (F = 92.438, $p$ = 8.0787E-38) and PC3 (F = 18.189, $p$ = 1.6942E-08) can be distinguished by region. Refinement can also be significantly distinguished between regions (F = 12.467, $p$ = 4.969e-06), driven by relatively thicker handaxes in the African sample, likely related to the much smaller sample sizes in the later African assemblages (see section 3.2.).

That shape is distinct between regions is confirmed by a MANOVA utilising the first 23 PCs (Wilks' λ = 0.7731, $p$ = 4.914E-35). Bonferroni-corrected post-hoc comparisons also confirm that each region is distinct from the other two (Table 2). These differences were tested further through a jack-knifed Linear Discriminant Analysis (LDA)—using the 23 PCs as the input—which correctly assigned 47.4% of handaxes to their region defined *a priori*, substantially above the rate of chance (33.3%).

A Levene's Test suggests variance on PC1 is not equal between regions (F = 4.0332, $p$ = 0.01798), driven by a greater consistency of shape in the African sample. This is the opposite pattern than would be expected in the event of a serial winnowing of shape variation as a result of repeated founder effects (cf. [25]). In order to explore this difference further, the intra-site and inter-site variances for each region are shown in Fig 5. The site-level violin plots and superimposed standard deviations for each region confirm that shape variance in general

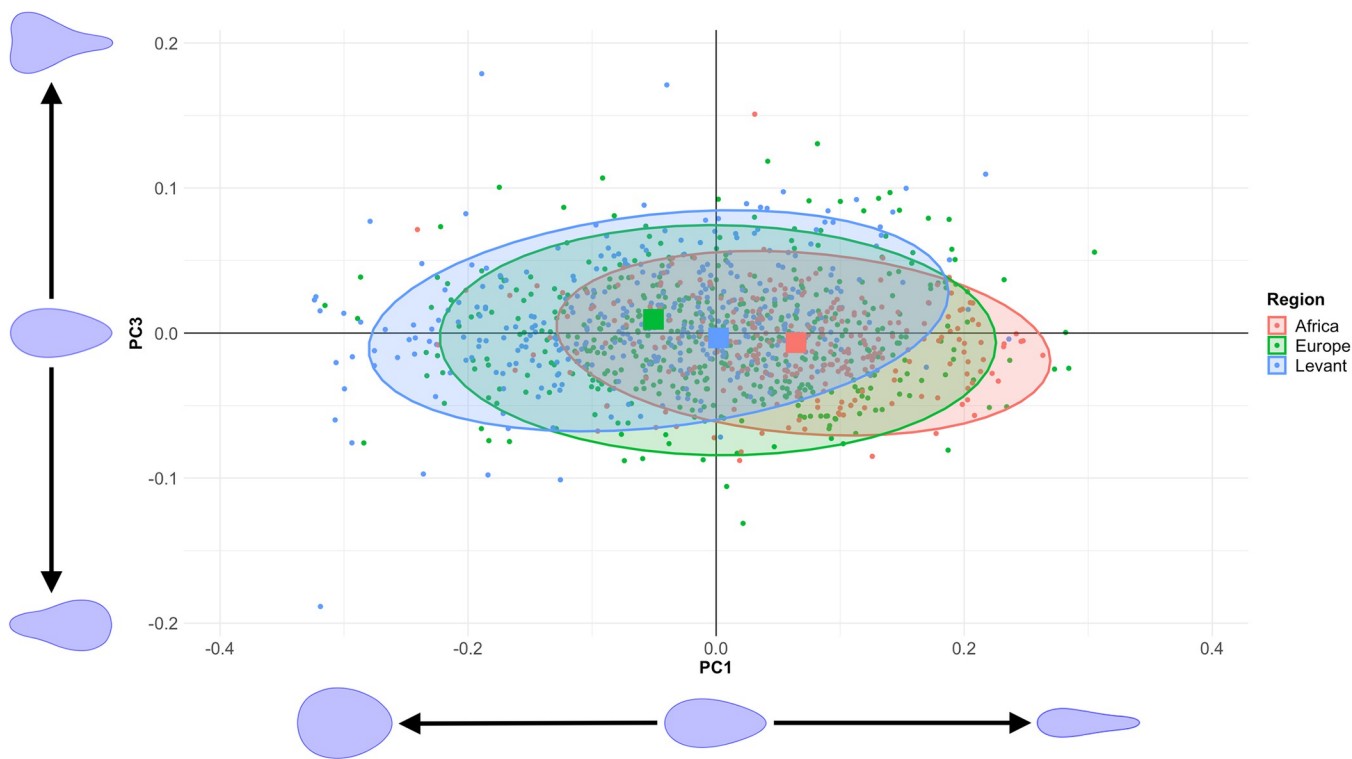

**Fig 4. Scatterplot of PC1 against PC3 from the EFA PCA.** Mean values for each region are denoted by the larger squares. The shape variance described by PC1 and PC3 when other PCs are held at 0 are displayed by each axis.

tends to be lower at African sites, but also that intra-site variance tends to be lower relative to variance in the wider African sample. This is confirmed by a Kruskal-Wallis Test into the standard deviations of site-level PC1 between regions (Kruskal-Wallis $\chi^2$ = 7.0044, $p$ = 0.03013). Intra- and inter-site variance appear to be more similar in Europe and the Levant. One possibility is that the smaller average size of site-level African samples is under-sampling total variability. However, the relatively lower variation in the African samples is repeated when controlling for sample size by using standard error, but the difference in intra-site variability between regions is no longer significant (Kruskal-Wallis $\chi^2$ = 2.5385, $p$ = 0.281).

Fig 6 shows the distribution of PC1 scores across time in each region, and suggests no obvious trends towards specific changes in shape over time. The one possible exception to this is within the Levant, whereby the earliest assemblages (from Gesher Benot Ya'aqov) are much more elongated and pointed than later sites in the region. This trend is only disrupted by the Tabun Eb handaxes from MIS9, which may reflect inter-assemblage differences in site use at

**Table 2. Bonferroni-corrected p-values from post-hoc MANOVA comparisons of shape between regions, based on the first 23 PCs.**

|  | Europe | Levant | Africa |
|---|---|---|---|
| **Europe** | – | 4.2502E-15* | 5.3131E-09* |
| **Levant** | – | – | 1.9442E-31* |

The results suggest highly significant differences between regions.

*Significant at α = .05.

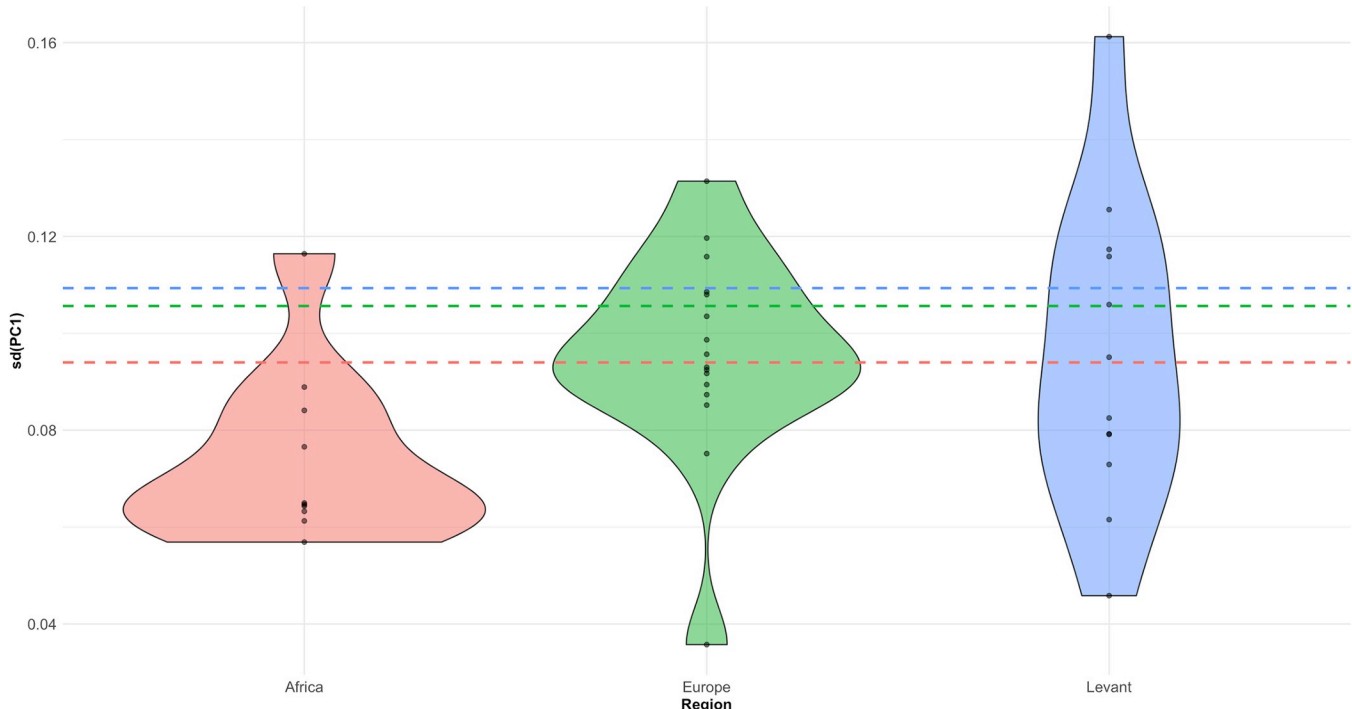

**Fig 5. Violin plots showing the site-level standard deviation in PC1 values within each region.** The standard deviation of artefacts from the whole region is displayed as dotted lines. The figure suggests that inter-site variance in shape may be larger than intra-site variance for the African samples, whereas it is relatively equal for Europe and the Levant.

Tabun [41]. The African samples may show a trend towards increasing inter-site variability of shape over time, but a wider range of Middle Pleistocene samples would be needed to test this further.

**3.1.2. Plan symmetry.** As shown in Fig 7, asymmetry across both main axes of the handaxe is extremely similar between regions, with very few clear differences. Kruskal-Wallis tests

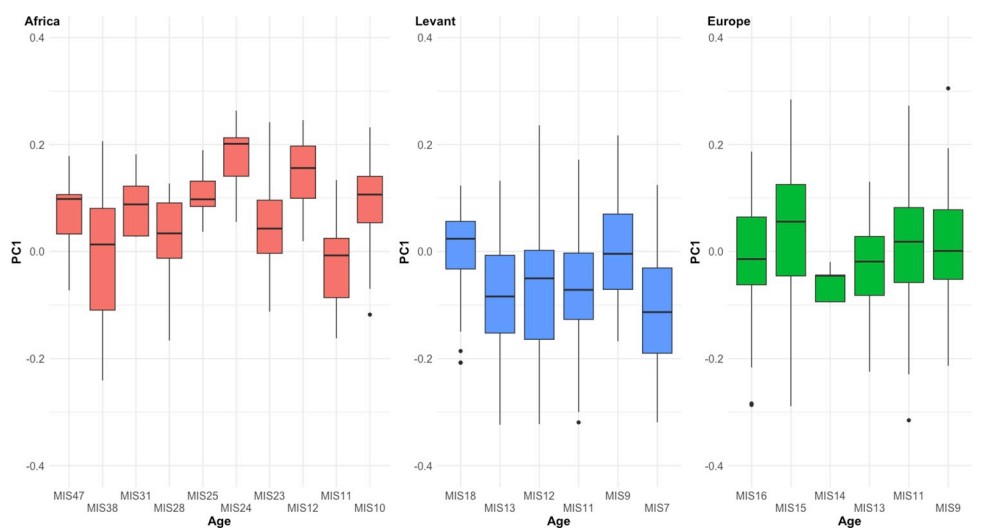

**Fig 6. Changes to PC1 over time in each region.** There are no clear trends in shape variability in any region.

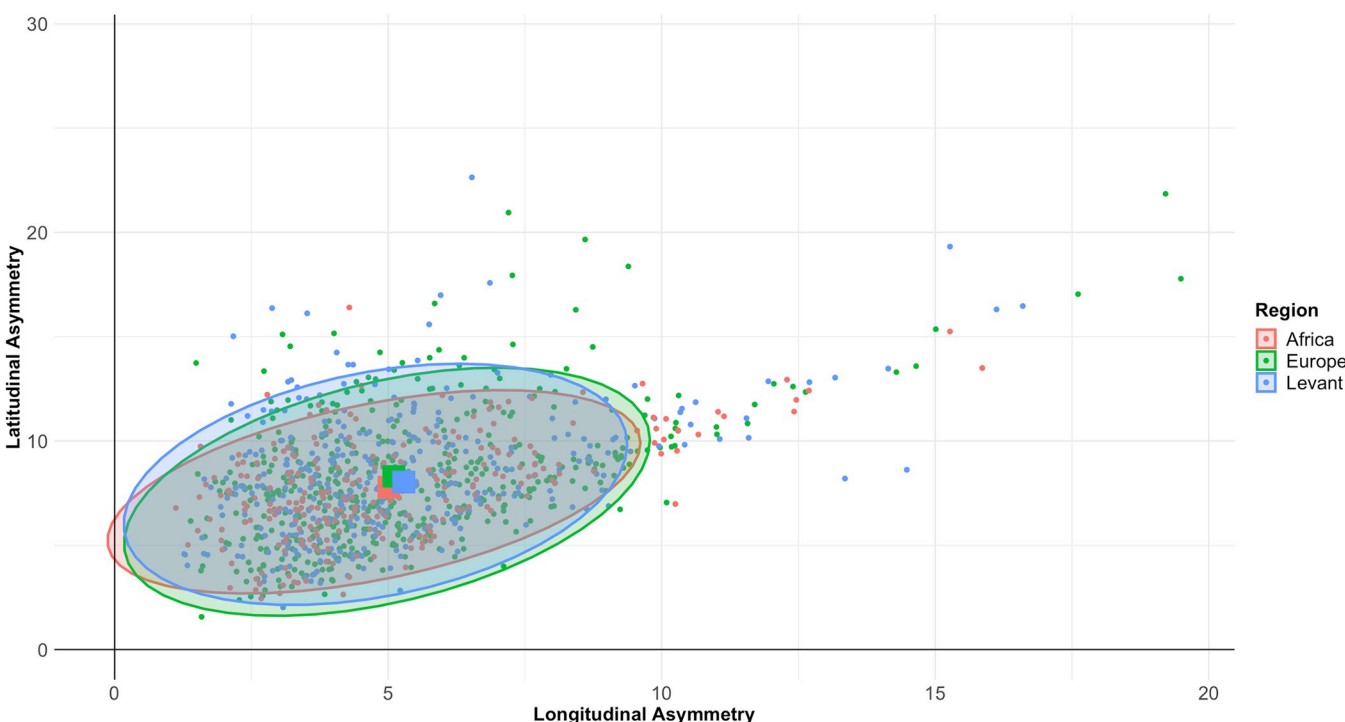

**Fig 7. Scatterplot of longitudinal against latitudinal asymmetry.** Mean values for each region are denoted by the larger squares. The results show no differences in plan symmetry between regions about either the length (longitudinal) or width (latitudinal) axes.

confirm that both median longitudinal (Kruskal-Wallis $\chi^2$ = 3.6525, $p$ = 0.161) and latitudinal (Kruskal-Wallis $\chi^2$ = 3.9751, $p$ = 0.137) asymmetry values do not vary significantly by region. The median absolute value of PC2 also did not significantly differ between regions (Kruskal-Wallis $\chi^2$ = 2.6271, $p$ = 0.26889). If plan symmetry is representative of cognitive ability (e.g. [78]), these data suggest no overall differences between hominins in each region. Furthermore, it is likely that plan symmetry is not subject to the same pressures that are driving differences in raw shape between regions, and variability is likely to be driven by local, site-specific influences.

There are no clear trends of longitudinal asymmetry over time in either the Levant or Europe, although there may be some increase in plan symmetry at the end of the Early Pleistocene in the African sample, interrupted by substantial inter-assemblage variability across Kariandusi and Olduvai Bed IV centred on MIS23 (Fig 8). The interruption to the broader trend may therefore be an artefact of averaging a number of sites to fit a single marine isotopic stage. Plan symmetry is, however, clearly lowest in the European sample at the very beginning of the European Acheulean, suggesting a temporary reduction in investment upon dispersal into the continent. The lowest plan symmetry values in the Levant in MIS7 are the result of the Nadaouiyeh A assemblage, which is the culmination of a longer-term trend towards reducing plan symmetry in that landscape [76,79,80], and may mirror potential trends towards reducing plan symmetry in all regions at the end of the Acheulean range.

## 3.2. Blank type

It is well known that prevailing blank types by region vary according to geological constraints on the availability of certain resources, especially volcanic rocks [35–37]. Data regarding the blank on which handaxes were produced were not collected for the present study, but

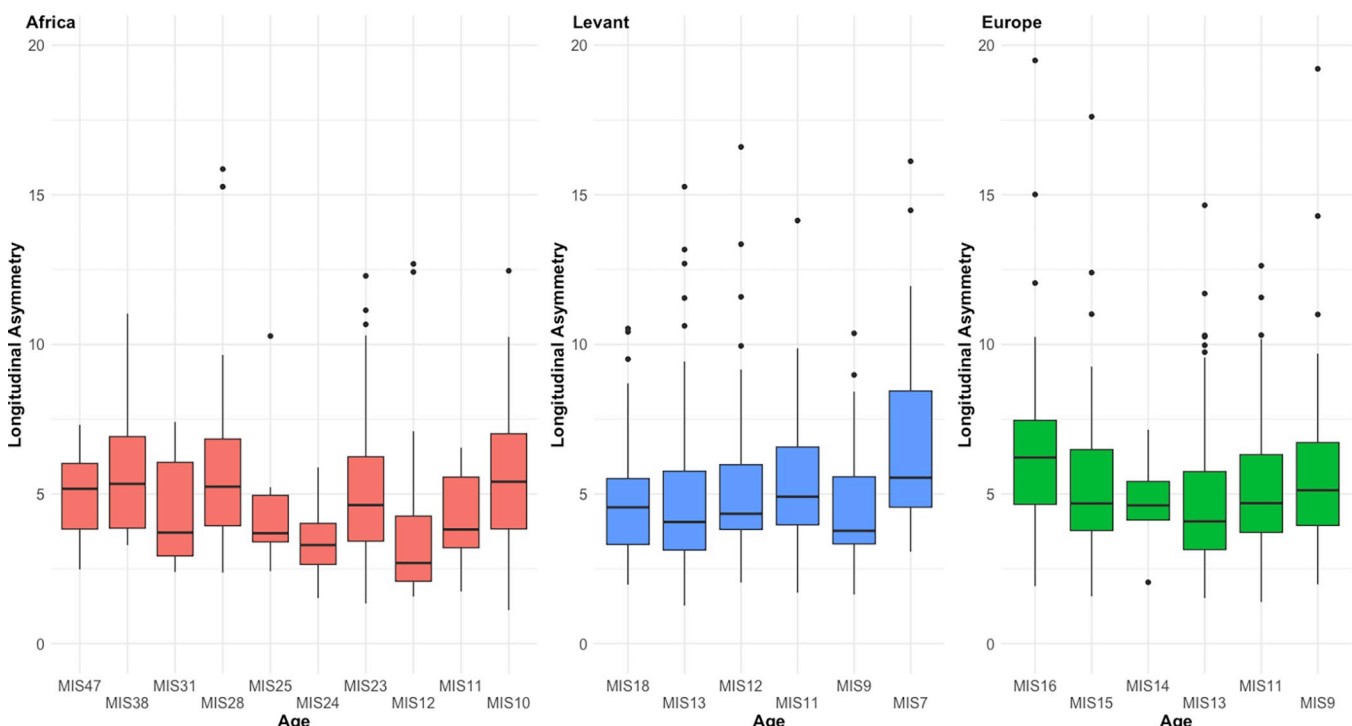

**Fig 8. Changes to longitudinal asymmetry over time in each region.** There are no obvious trends in the Levantine or European samples, although there may be an increase in plan symmetry at the end of the Early Pleistocene in Africa.

independent data were available for 291 specimens, displayed in Table 3. Nonetheless, the 90 handaxes from the three levels of Gesher Benot Ya'aqov (hereafter GBY)—all made on flakes —are excluded from further analysis unless explicitly stated otherwise, so as to not bias the results towards a single site and region.

A MANOVA utilising the first 23 PCs suggests that there are no significant differences in shape by blank type for the remaining 201 specimens (Wilks' λ = .7678, $p$ = .341). This is repeated even when the data from GBY are included (Wilks' λ = .8126, $p$ = .1197). Fig 9 shows the distribution of scores for the first three principal components according to known blank type, and shows very little difference in median value by blank. The strongest exception to this pattern is the elevated PC1 values on slab and flake blanks when compared to cobble blanks (F = 3.811, $p$ = .024, which is almost certainly due to an overrepresentation of cobbles in the European and Levantine samples and an overrepresentation of flakes and slabs in the African samples. A Levene's Test indicates no significant differences in PC1 variance by blank type (F = 1.7818, $p$ = 0.1489), which further suggests any lower intra-site variance in the African samples is not being driven by any intrinsic feature of flake blanks.

**Table 3. Blank breakdown by region.**

| Blank | Africa | Levant | Europe | Total |
|---|---|---|---|---|
| Cobble | 2 | 26 | 26 | **54** |
| Flake | 74 | 105 | 40 | **219** |
| Slab | 18 | – | – | **18** |
| Unknown | 170 | 224 | 412 | **806** |
| Total | **264** | **355** | **478** | **1097** |

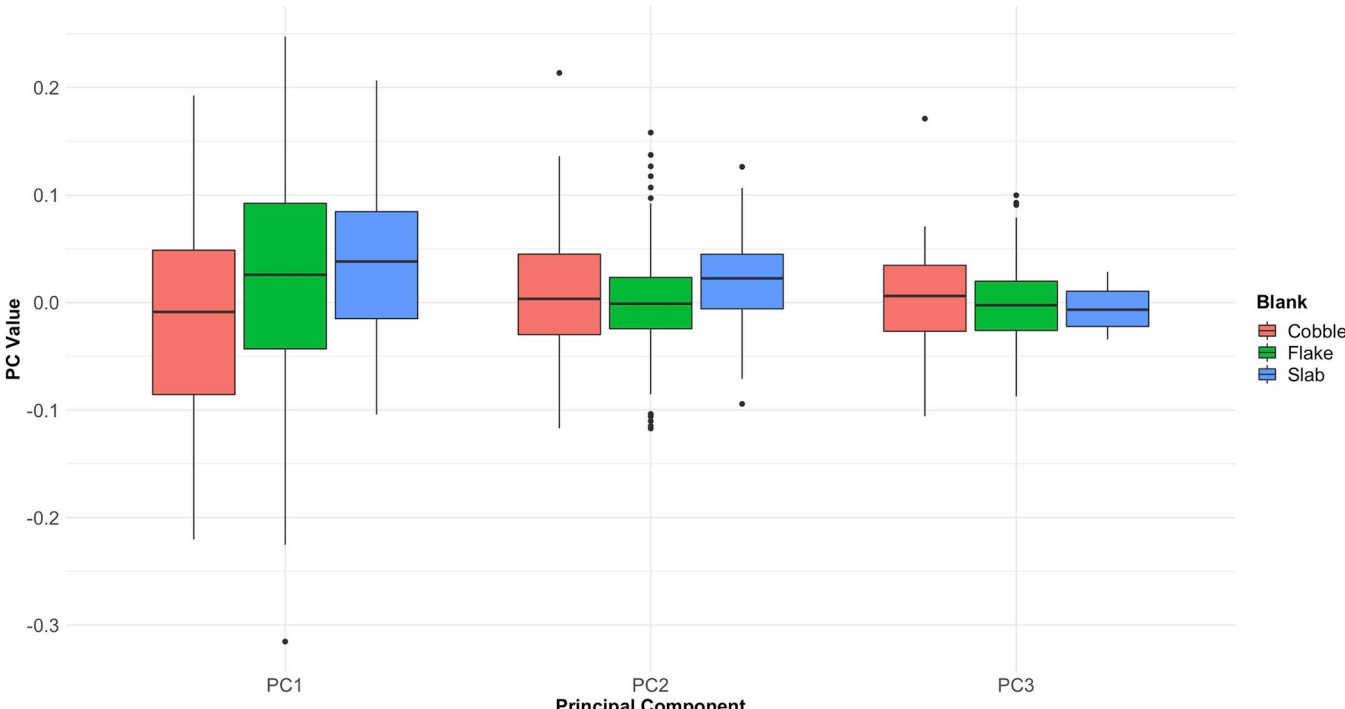

**Fig 9. Boxplot showing values for the first three PCs derived from the EFA PCA by blank type.** The results indicate the existence of some differences, but these are likely driven by the differential selection of different blank types between regions.

In contrast, however, the significant difference in refinement values by blank type (F = 5.8976, $p$ = 0.001247) cannot be explained by broader regional trends. This is because cobble artefacts are the thickest specimens in the sample, and yet Europe and the Levant display the lowest overall refinement values, indicating the presence of the thinnest specimens. It seems likely that this finding is instead an artefact of time in the analysis, given the African samples cover a much broader range of time (while underrepresenting later assemblages), and the earliest of these artefacts are much thicker than later ones (Fig 10). This difference reduces but remains when only looking at artefacts definitively younger than 1 Ma (F = 3.3697, $p$ = 0.03524), however the sample of African handaxes was not large enough to make a direct Middle Pleistocene comparison to the European and Levantine samples. Given the limited information from the later African samples in Fig 10, it seems likely that the difference would disappear if such a test was possible.

A further attempt to control for the impacts of blank type was to focus only on handaxes made from specific blanks within each region. The sample contained a relatively low number of specimens made on slabs in total (n = 17), and a rarity of African specimens produced on cobbles, and thus this was only possible using large flake blanks. As shown in Fig 11, restricting our analysis only to flake blanks replicates the regional patterns seen in the wider dataset. This includes variation in PC1 values by region (Kruskal-Wallis $\chi^2$ = 24.447, $p$ = 4.9134E-06), with the highest values in African specimens and lowest in Levantine specimens. Similar tests revealed no differences in PC2 by region (Kruskal-Wallis $\chi^2$ = 2.5289, $p$ = 0.2824), but there was a significant difference in PC3 (Kruskal-Wallis $\chi^2$ = 10.153, $p$ = 0.006241), with Levantine specimens having the highest values. Refinement values can also be differentiated by region when only looking at artefacts made on flakes (F = 31.239, $p$ = 5.745e-09), driven again by thicker specimens from the African Early Pleistocene.

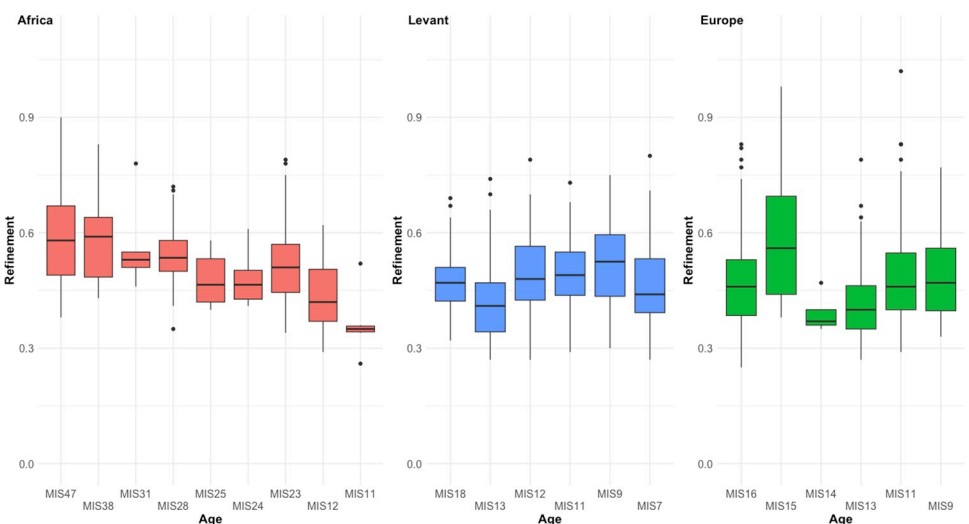

**Fig 10. Changes to refinement over time within each region.** The African sample shows a clear trend towards reducing refinement over time that is not seen in the other two samples.

Fig 12 shows that differences in PC1 are also present between Europe and the Levant when focusing only on handaxes made on cobble blanks, with this difference also significant (Mann-Whitney U = 454, p = 0.0337). Differences in refinement, however, did not reach significance in this subsample (U = 417.5, p = 0.1479). Overall, these results indicate that blank type is not

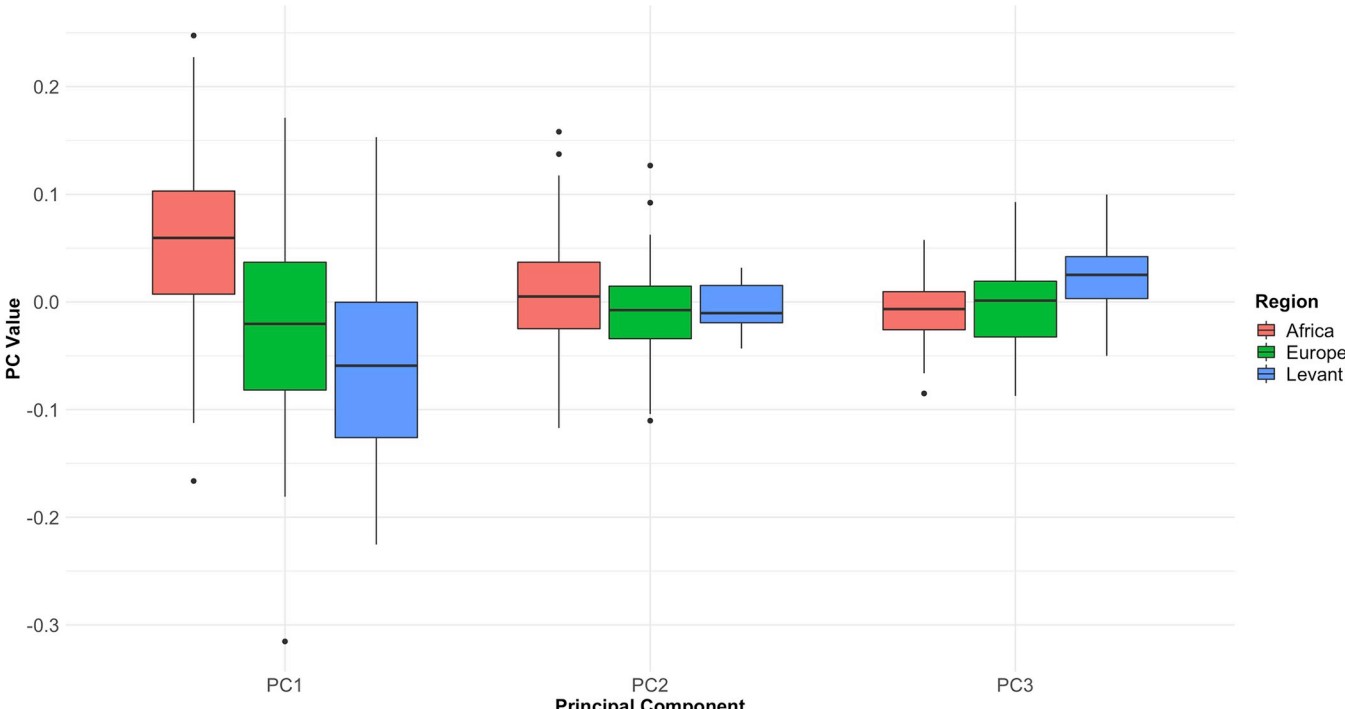

**Fig 11. Boxplot showing values for the first three PCs derived from the EFA PCA, when only looking at artefacts made on flake blanks.** The results clearly replicate the findings from the wider dataset, suggesting blank type is not driving the differences between region.

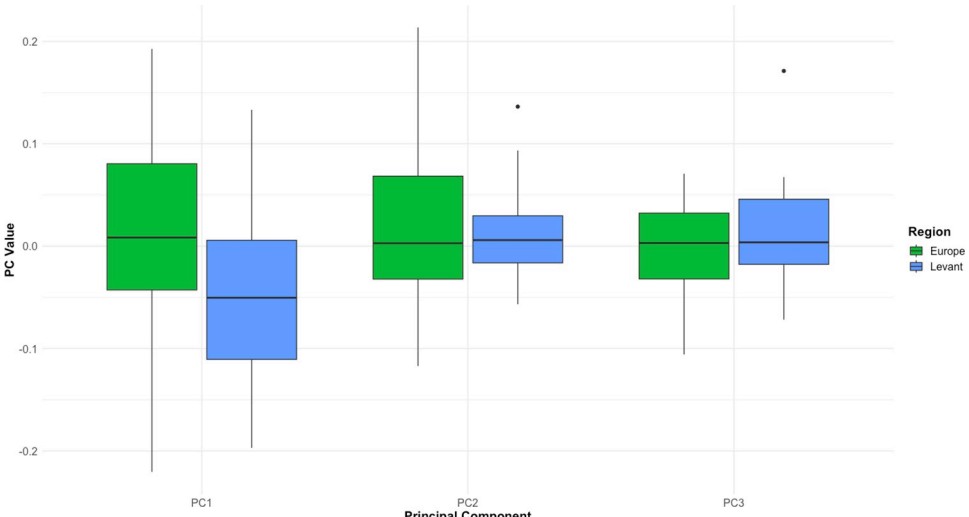

**Fig 12. Boxplot showing values for the first three PCs derived from the EFA PCA, when only looking at artefacts made on cobble blanks.** As there are only two confirmed pieces made on cobbles in the African sample, the comparison was limited to Europe and the Levant.

driving the differences in shape between regions, as different forms were being produced even when blank is held constant.

### 3.3. Raw material

A similar analysis was run when individual raw materials were represented by $\geq 15$ specimens from across at least two assemblages. Following previous literature, multiple igneous rock types were subsumed into the wider category of "lava" because of differences in the detail of information available for different sample sources. Obsidian was excluded from this category and all others because its crystalline structure is substantially different from other lava types, and more similar to chert [53]. Flint and chert were also kept separate because both groups had large enough individual samples. The resulting frequencies by region are shown in Table 4.

A MANOVA on the first 23 PCs revealed that shape did vary significantly by raw material (Wilks' $\lambda$ = .8239, $p$ = 1.789E-07). These differences on the first three PCs are illustrated in Fig 13, with PC1 showing the clearest differences (F = 23.4, $p$ = 1.4901E-18). However, post-hoc MANOVA tests suggest the only material contrasts to reach significance for the first 23 PCs were those between lava and millstone, lava and flint, and quartzite and flint (Bonferroni-corrected $p < .05$). This pattern is not unexpected, given lava and quartzite handaxes are

**Table 4. Raw material breakdown by region.**

| Raw Material | Africa | Levant | Europe | Total |
|---|---|---|---|---|
| Chert | 26 | 12 | 38 | 76 |
| Flint | — | 253 | 371 | 624 |
| Lava | 114 | 90 | — | 204 |
| Quartz/Quartzite | 90 | — | — | 90 |
| Millstone | — | — | 51 | 51 |
| Unknown | 34 | — | 18 | 52 |
| Total | 264 | 355 | 478 | 1097 |

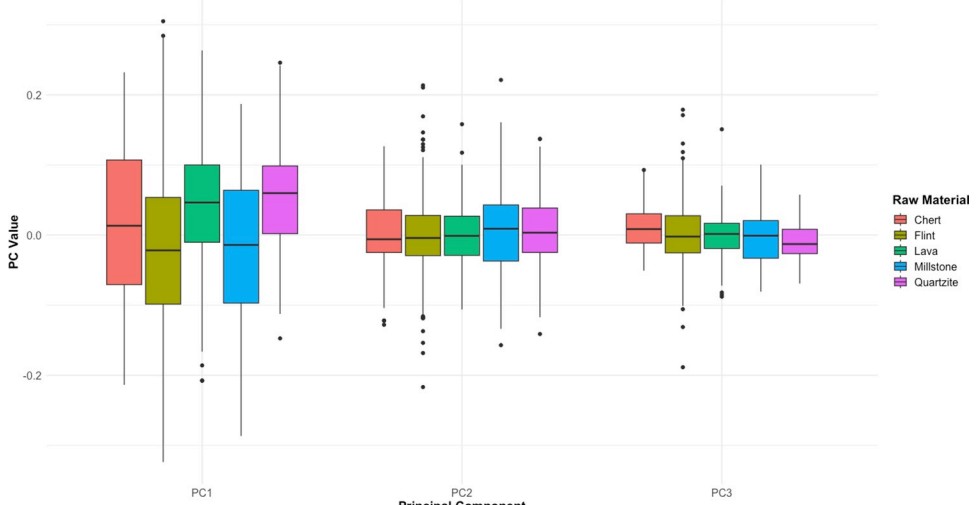

**Fig 13. Boxplot showing values for the first three PCs derived from the EFA PCA, according to raw material.**
Differences exist between raw materials, but these are likely to reflect the differential availability of individual raw
materials between regions.

overwhelmingly from Africa, while flint handaxes are exclusively from Europe and the Levant,
and millstone exclusively from Europe. Chert, the only material present in all three regions,
did not come close to being distinguished from any of the other rock types in these post-hoc
tests (all $p > .05$). Further still, a single raw material in flint covered almost the entire range of
possible shape values for PC1-3, undermining the suggestion that specific raw materials have
biased shape results.

Refinement by raw material presents a similar difficulty as blank type, given the bias
towards the Early Pleistocene in the African sample. As such, significant differences do emerge
(F = 3.5096, $p = 0.004775$), driven by thicker artefacts made on quartzite and lava. Nonetheless,
the relatively thinner artefacts in the later (and smaller) African samples implies that this is
unlikely to be an intrinsic property of these materials. Chert displays the lowest refinement val-
ues of all raw materials found at more than one site, further suggesting that differences
between regions are not related to differences in material availability and selection by region.

### 3.4. Allometry

Two allometric relationships are examined to explore the effect of resharpening on handaxe
form, namely changes to overall handaxe shape and changes to handaxe thickness throughout
knapping. We present each aspect separately.

**3.4.1. Changes to plan shape.** Throughout the entire handaxe sample, the correlation
between tip length and PC1 (Fig 14) is highly significant (r = .642, $p = 1.2889E-128$), support-
ing an allometric relationship between handaxe shape and size. However, when this relation-
ship is broken down into site-specific coefficients (a total of 38 assemblages with n $\geq$ 15), there
is no difference in median correlation coefficient by region (Kruskal-Wallis $\chi^2$ = .019, p =
.991), suggesting the differences in PC1 by region cannot simply be explained by stronger
adherence to this trend in any one region. Indeed, the relationship between site-level mean
PC1 and site-level Pearson's coefficients between PC1 and Tip Length suggests very little rela-
tionship between the two (r = 0.119, $p = 0.477$), as shown in Fig 15.

Nonetheless, it may be argued that the relationship between PC1 and Tip Length cannot, in
itself, be used to measure resharpening in the way originally conceptualised by McPherron

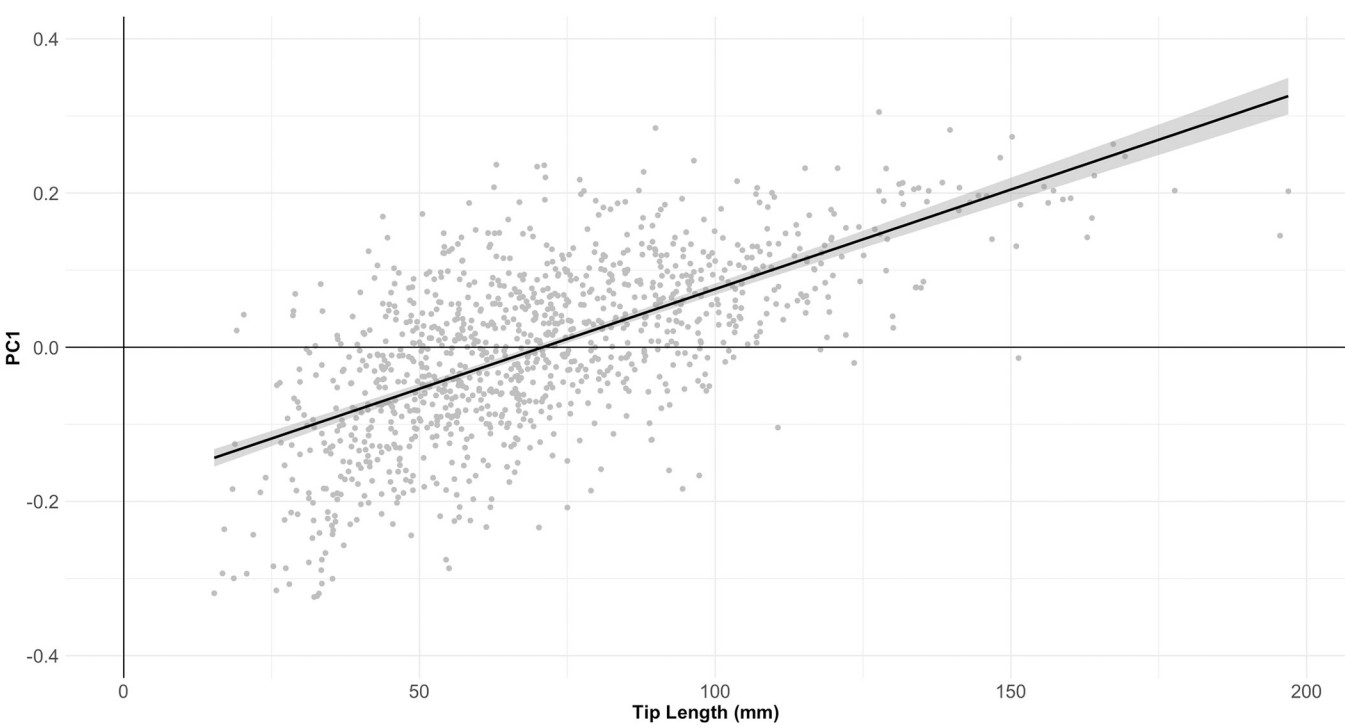

**Fig 14. Scatterplot of Tip Length against PC1 for the entire sample.** The results indicate a clear positive relationship.

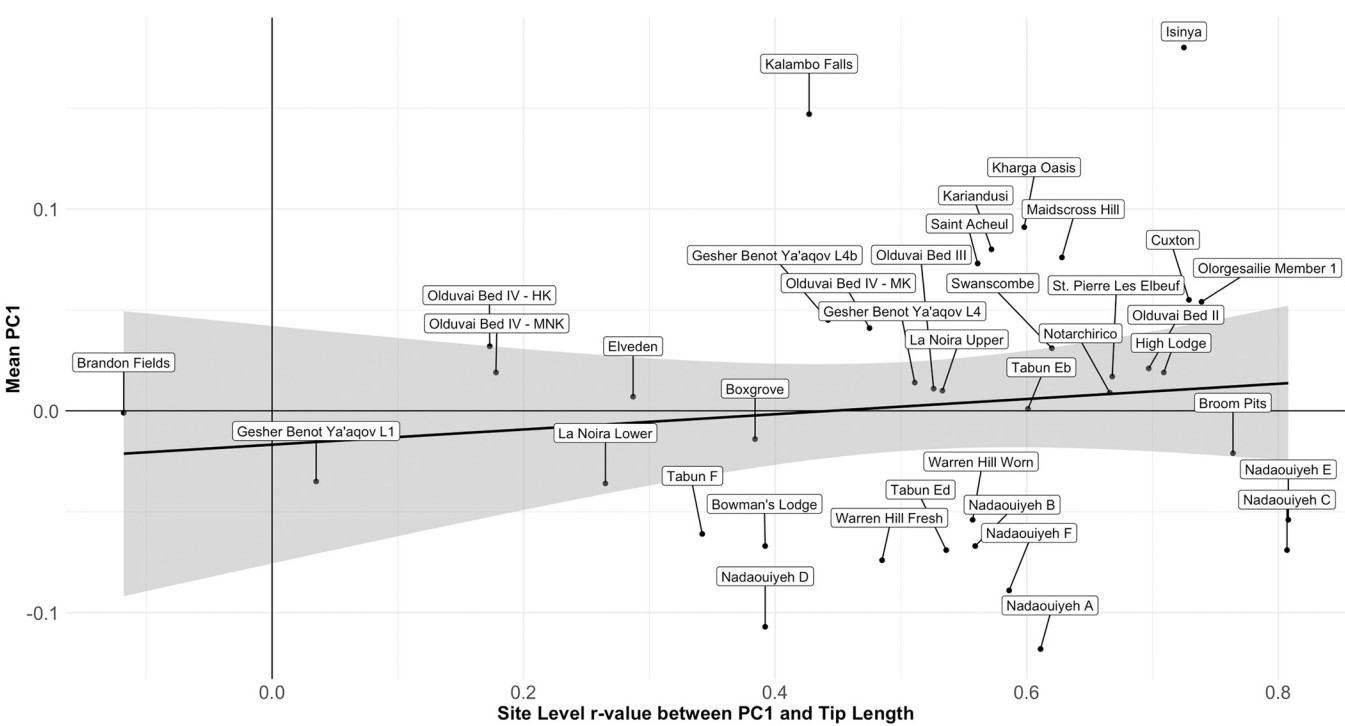

**Fig 15. Scatterplot of site-level mean PC1 values against site-level Pearson's r-values between PC1 and Tip Length.** The results indicate that site-level changes to PC1 as tip length is reduced are not driving overall differences in shape between sites.

[38,39], which focused only on Roe's [44,45] assessment of Tip Shape (Width at 1/5 of Length over width at 4/5 of Length). As mentioned previously, PC1 variance is almost completely explained by Elongation and Tip Shape together, but is largely represented by the former. As such, PC1 is not truly independent of Tip Length as a predictive variable, given it is highly likely that handaxes with longer tips will be longer relative to their width, regardless of resharpening. Indeed, the relationship between Tip Length and Elongation is highly significant (r = -.602, *p* = 4.3161E-109) and remarkably similar to the relationship between Tip Length and PC1.

When controlling for Elongation in the relationship between Tip Length and PC1, the result remains significant, but is substantially weakened ($r_{partial}$ = .281, *p* = 2.4049E-21), with a large amount of variability in the site-level relationships. Indeed, a number of assemblages do not present a positive relationship when controlling for Elongation, which may suggest the functional assumptions of McPherron's [38–40,42] resharpening relationship are not always met, or that resharpening does not always follow the same trajectory. In turn, when these site-level partial r-values are themselves correlated with site-level mean PC1 values—again controlling for the influence Elongation on PC1 ($r_{partial}$ = .422, *p* = .009). This finding shows that sites with the most pointed average shapes also show the strongest resharpening coefficients in which the tip becomes more rounded throughout reduction, rather than being a product of the earliest stages of reduction. Taken together, these data suggest that resharpening—in the ways measured and discussed by 8, 38–42]—plays only a minor role in driving variation between sites and, by extension, regions. It can only perhaps be suggested that sites with a more pointed handaxe distribution (particularly African assemblages) have been subject to greater levels of resharpening, counter to the pattern expected for regional shape variation being driven by differences in the extent of resharpening.

**3.4.2. Changes to thickness.** As a preliminary test of the functional implications behind the allometric changes to shape, we looked for variables which may reflect a different knapping trajectory, specifically one primarily related to the production of usable flakes rather than a usable cutting edge. Changes to refinement throughout the knapping process may provide an indirect test of this notion, because handaxe sharpness (and thus cutting potential) is dependent on the cutting angle at the lateral edges [81], whereby relatively thinner handaxes have the potential for smaller cutting angles [53]. One hypothesis is that handaxes should show a parabolic relationship between reduction stage and refinement, with handaxes initially becoming thinner as they are reduced, and subsequently becoming thicker as they progress towards exhaustion [40,42,43]. Under these circumstances, we would expect sites with the least and greatest emphasis on resharpening to have the greatest mean refinement values (i.e. the longest and shortest specimens, respectively, are thickest relative to width), and sites with intermediate levels of resharpening to have the lowest (i.e. that the shortest specimens are thinnest relative to their width). McPherron [40–42] reports that there will either be a decrease in refinement throughout knapping, or that it will stay relatively constant as width and thickness are modified at a similar rate.

The results are consistent with the last of these possibilities, with no linear relationship between the resharpening coefficient and mean refinement at each site (r = .090, *p* = 0.5966), and no additional evidence for a U-shaped curve (Fig 16). It is true that the lowest mean refinement values (with the thinnest handaxes relative to width) are seen at two sites with intermediate resharpening coefficients, but other sites with similar resharpening relationships have much higher values. There is a general spread of refinement values throughout the range of resharpening coefficients.

As a consequence of this neutral relationship between resharpening (sensu [38,39]) and mean refinement, we tested for additional changes to core morphology during the knapping

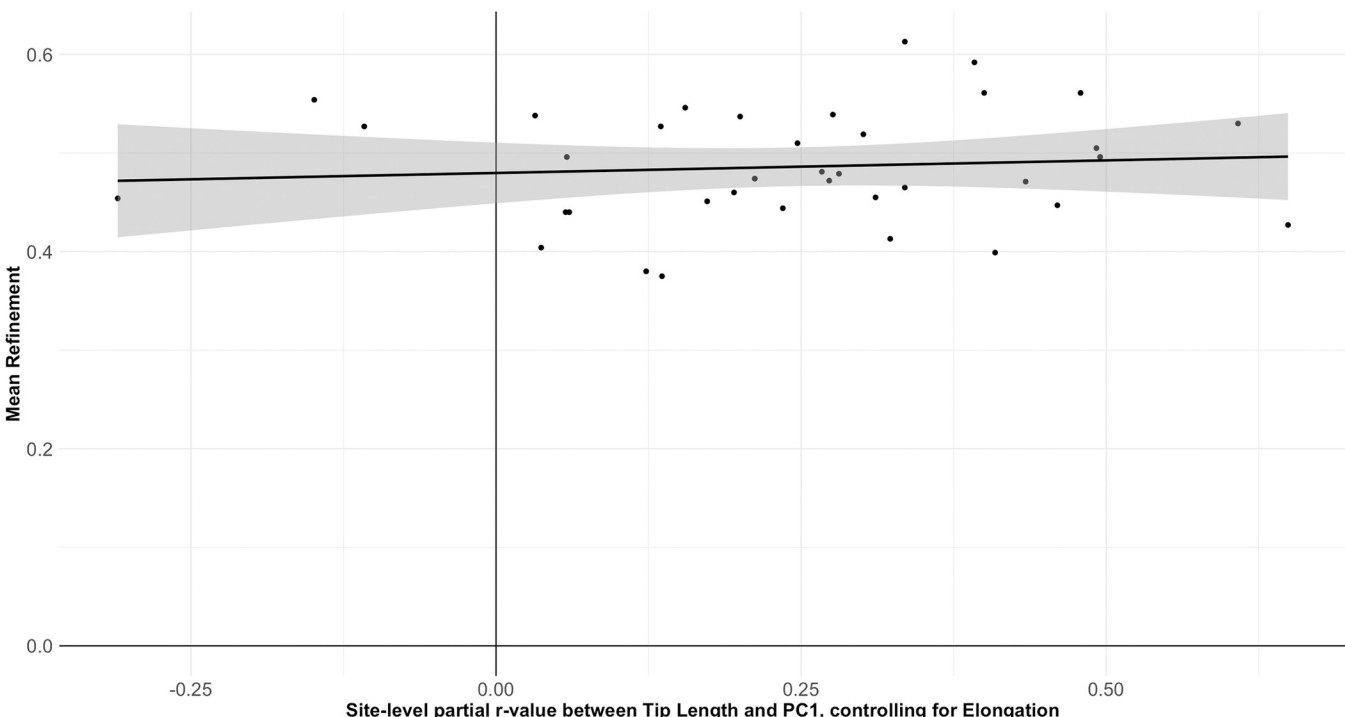

**Fig 16. Scatterplot of site-level mean Refinement values against site-level relationships between Tip Length and PC1, controlling for Elongation.** There is clearly no relationship between the variables, suggesting that changes to overall shape attributable to resharpening are not associated with changes to refinement.

sequence. In particular, we tested whether cores reduced for the production of flakes show an increase in their thickness relative to width, because there is no pressure to maintain low cutting angles, while the use of lateral edges as a platform for flake production may reduce the width axis faster than the thickness axis. As a semi-independent metric for this relationship, we looked at the site-level correlation between Length and Refinement, given no "tip" portion should necessarily exist for other core forms. Data collated for an unrelated reduction experiment seems to confirm that cores show a consistent negative relationship between length and refinement, such that specimens become relatively thicker as their length is reduced [82].

In contrast, there is no overall relationship exists in the wider handaxe sample (r = 0.0187, $p$ = 0.5427), but there is a large amount of variation between sites and regions. African handaxes show a weak negative relationship between these variables (r = -.289, $p$ = 1.109E-05), although it is clearly absent from both the Levantine (r = .009, $p$ = 0.8726) and European (r = .020, $p$ = 0.6557) samples. Site-level relationships between Length and Refinement were then themselves correlated with the site-level partial coefficients between Tip Length and PC1 controlling for Elongation (as a coefficient of resharpening). Interestingly, the relationship is extremely close to reaching significance (r = .302, $p$ = .069). As shown in Fig 17, Swanscombe and especially Bowman's Lodge are notable outlying values, with the latter falling at the very end of variation for site-level resharpening coefficients. If the test is re-run to the exclusion of Bowman's Lodge, the relationship does reach significance (r = .413, p = .012), illustrating its disproportionate skew on the overall relationship.

That there is a relationship between these two site-level coefficients may be suggestive of a continuum of possible handaxe reduction strategies. This may extend from a strategy more focused on the production and maintenance of a sharp cutting edge (i.e. strong evidence of tip

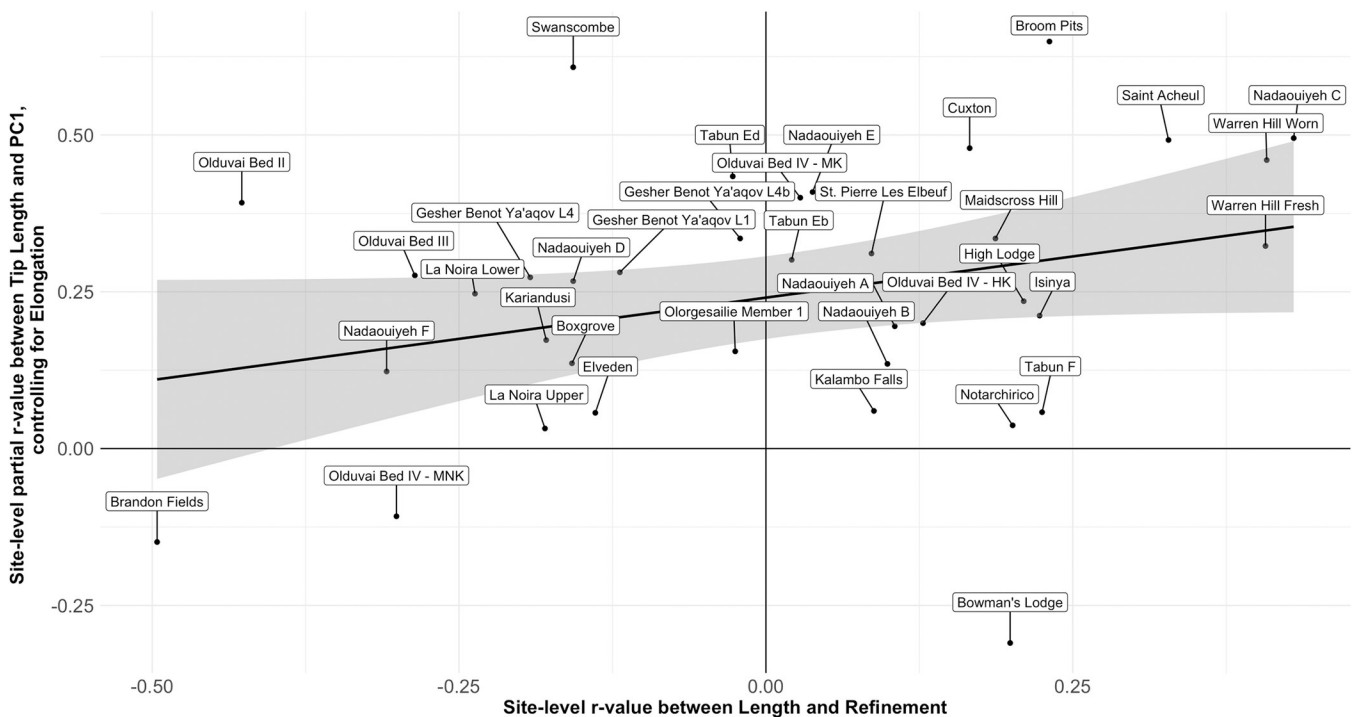

**Fig 17. Scatterplot of site-level relationships between Length and Refinement against site-level relationships between Tip Length and PC1, controlling for Elongation.** We wish to emphasise that the position of sites along either axis is not a reflection of planform shape or relative thickness, but how each of these change across the knapping sequence. The results may indicate the presence of a continuum of possible site-level reduction strategies, ranging from one more focused on the maintenance of tip shape and the production of flakes on the left side of the distribution, to one more focused on the production and maintenance of a sharp cutting edge on the right side of the distribution.

rounding on one hand and thinning of the artefact on the other as the handaxe is reduced), to one more focused on the maintenance of tip shape and the production of flakes (limited evidence of changes to tip shape throughout reduction, but a trend in which handaxes become thicker relative to their width as knapping progresses). It is important to note that specimens focused on the production and maintenance of a cutting edge will still produce usable flakes, and those adapted to flake release would still have usable cutting edges: the potential position along such a spectrum would simply highlight the greater optimisation of handaxe forms for one objective over the other. Assemblages from each continental region are spread across this spectrum, suggesting a shared flexibility in handaxe deployment that is unrelated to regional trends of shape.

## 4. Discussion

The main results presented in the paper are summarised in Table 5. While it is important to emphasise the sampling biases within and between regions, the clear patterning of shape between regions is consistent with findings from a number of previous investigations (e.g. [21,22,24,83]). This is clearly not explainable by the differences in raw material availability and selection that characterise the different regions included within the three main subsamples. Alongside these differences in average shape, there appears to be lower shape variation in the African sample, with intra-site variability in particular lower than variability in the whole sample. These differences are particularly striking given the more palimpsestic nature of certain sites included in the African sample (especially Olduvai Beds II and III). In contrast, Europe

**Table 5. Summary of the main findings presented in this study.**

| Finding | Conclusion |
|---------|-----------|
| 1 | When controlling for size, average handaxe shape is significantly different between regions |
| 2 | Differences in shape between regions cannot be explained by different properties of the different blank types and raw materials being used |
| 3 | Variation in handaxe shape is lower in the African sample than in the European or Levantine samples |
| 4 | African handaxes may show lower intra-site shape variability relative to the wider regional sample than those from Europe or the Levant, although this may be an artefact of sample sizes |
| 5 | There is a trend towards handaxes becoming thinner on average (and possibly more symmetrical in plan view) in the African samples (from the Early Pleistocene into the Middle Pleistocene) that is not seen in the Middle Pleistocene of the Levant or Europe |
| 6 | Measurement of resharpening in geometric morphometric investigations of handaxes must control for elongation |
| 7 | Differences in shape between region cannot be explained by differences in the extent of resharpening |
| 8 | Changes to shape attributed to resharpening are not associated with changes to handaxe refinement |
| 9 | Individual sites display a continuum of deployed reduction strategies, ranging from those more focused on the production and maintenance of a sharp cutting edge, to one more focused on the maintenance of tip shape and the production of flakes |
| 10 | Where sites fall on this continuum of reduction strategies is not patterned by region |

and the Levant possess a number of assemblages related to very short periods of time (especially Boxgrove—which may represent only 100 years of time and displays the lowest variability of all sites, representing a significant outlier for European sites—and individual levels of Gesher Benot Ya'aqov at the lower end of the Levantine distribution). Nonetheless, it is also true that the European and later Levantine sites have larger temporal brackets that derive from relative dating using Marine Isotopic Stages, and may therefore also reflect accumulation of artefacts over long periods of time.

With regards to allometry, we confirm the results of other 2D geometric morphometric investigations that PC1 is typically strongly associated with the length and especially tip length of the handaxe (e.g. [8,48,60]). This supports a growing body of literature that handaxe shape does not scale isometrically with size, regardless of what is driving this size variation (e.g. [38,39,84,85]). At the same time, we find that a simple relationship between size and principal components in Geometric Morphometric studies of Acheulean handaxes is not enough to conclude that resharpening is driving changes in shape. In these contexts, handaxe elongation is a clearly confounding variable, because it is the largest single determinant of PC1 variation, and therefore inevitably the two variables covary. We suggest that controlling for this variable allows PC1 to take on a better approximation of Roe's [44,45] tip shape, that was originally used by McPherron [38,39] to examine resharpening relationships. The lack of concordance between tip-length and PC1 relationships with more objective measures of reduction (e.g. Scar Density Indices) has led many to abandon the meaning behind more easily measured allometric trends [e.g. 43]. Nonetheless, our methodological clarification appears to be effective at detecting variation in resharpening relationships at individual sites, and highlights that resharpening cannot explain the differences in average shape between sites and between regions. This also highlights that the method has high utility for exploring changes across reduction in future 2D approaches when there is no access to more expensive and time-intensive 3D methodologies. These approaches should be directly complimentary in studies of lithic variability across the Palaeolithic.

Our results make it clear that different aspects of handaxe morphology (in particular shape, plan symmetry, and refinement) do not vary at the same spatial scales. While variability is likely introduced for all three variables according to site-level patterns of blank selection, raw

material use, and resharpening, particularly in the European and Levantine samples, it is only shape that can be clearly separated by continental region. Refinement may also show differences between regions, but this is likely an artefact of changes over time, given the bias towards Early Pleistocene specimens in Africa, and a trend towards reduced artefact thickness over time in the continent. The increased inclusion of earlier specimens is therefore likely to be biasing comparisons between regions, as the Levantine and European specimens exclusively date to the Middle Pleistocene. Instead, refinement, alongside plan symmetry, may be associated with site-level differences in handaxe deployment, creating the continuum of reduction strategies suggested in Fig 16. This accords with the conclusions of McNabb and Cole [86] for short-term fluctuations in handaxe plan symmetry from different data. Fig 18 summarises these differential influences on handaxe morphology at different scales, which we discuss separately below.

## 4.1. Regional drivers of handaxe variability

Handaxe shape is particularly interesting for its variation at both very local, and very broad spatial scales (Fig 18). The inter-site variability in shape within regions shown in Fig 19 highlights the importance of local influences on shape, perhaps related to the influences of blank type, raw materials, and resharpening. This includes substantial changes within extremely short periods of geological time, such as between levels at GBY, despite their manufacture according to similar technological schemes [57]. Nonetheless, at higher spatial scales, we have presented clear average differences between the continental regions of eastern Africa, the Levant and (northwest Europe). Work elsewhere on the European record also suggests an internal spatial patterning of handaxe morphology, such as between the Atlantic basin of western France and northern Spain, central France and southern England, and eastern France [51,52,87].

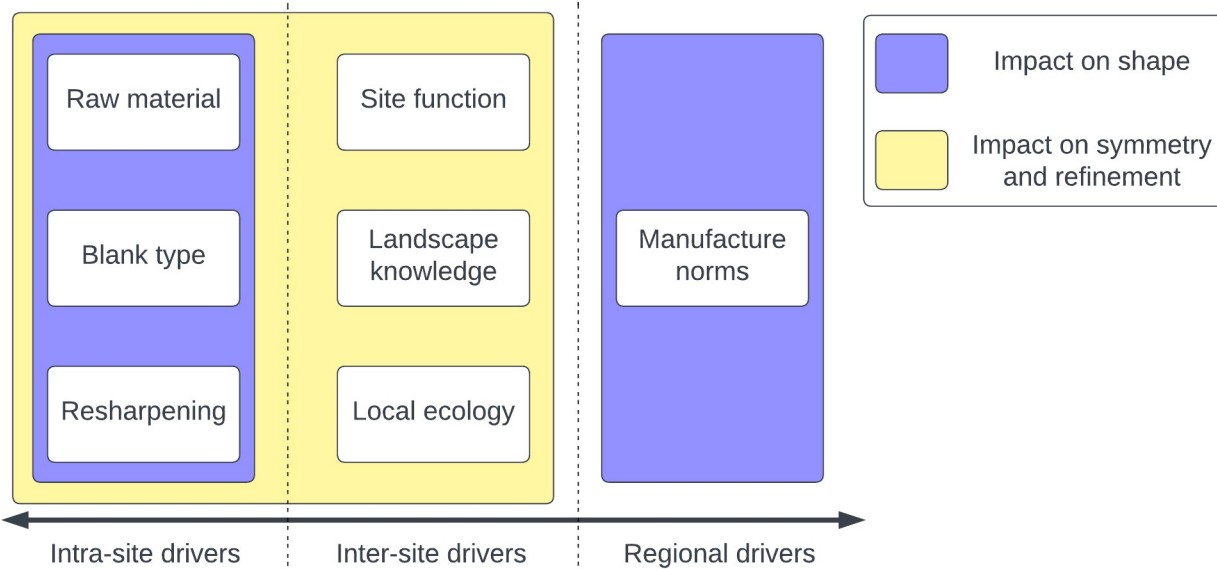

**Fig 18. Drivers of handaxe variation at different spatial scales, separated by their influence on different aspects of handaxe morphology.** Random fluctuations in shape are likely to be emerge as a result of extrinsic factors at individual sites, but broader scale patterning is likely to relate to regional norms of manufacture. Plan symmetry and refinement may share these intra-site fluctuations, but otherwise are likely to reflect patterns of local ecology, landscape knowledge, and site function.

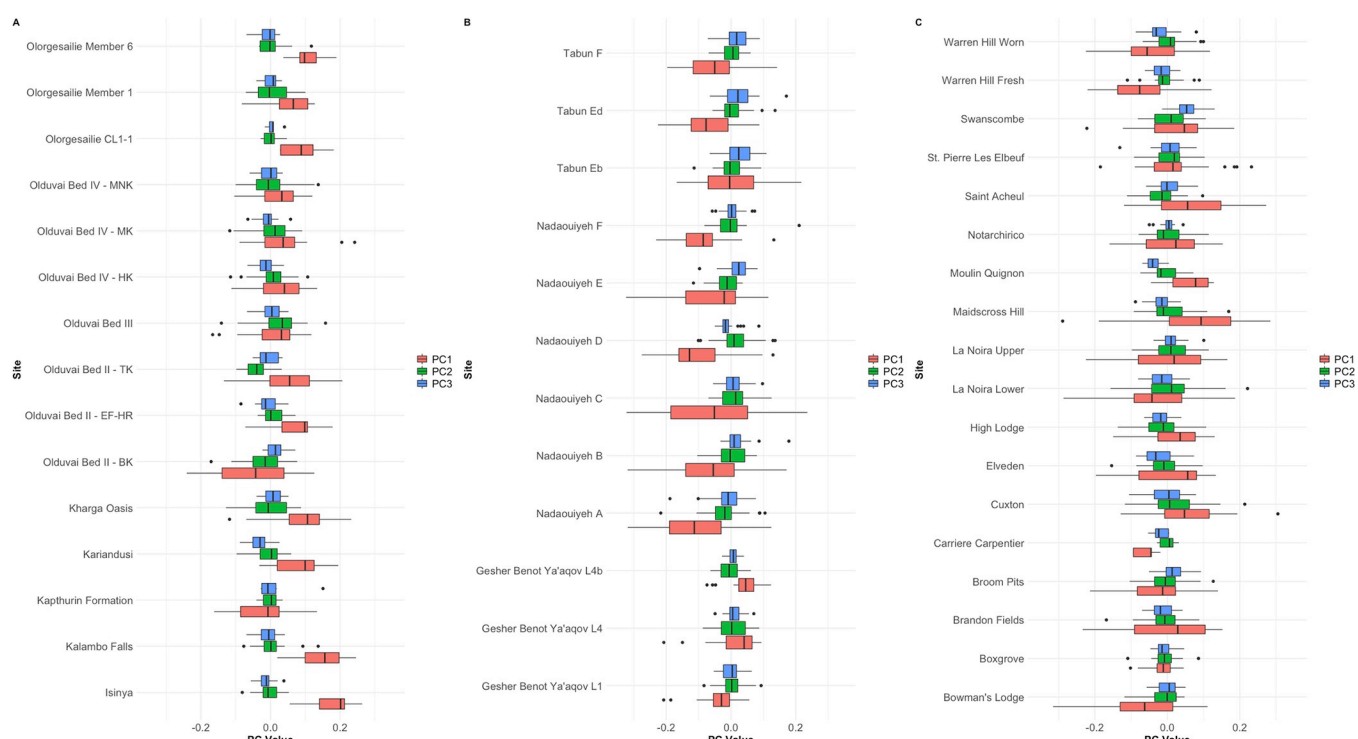

**Fig 19. Boxplot showing values for the first three PCs derived from the EFA PCA, according to site.** A = East African Sites, B = Levantine Sites, C = European Sites. The results highlight the substantial variability that exists between sites within individual regions.

According to these findings, it appears likely that a) different norms of handaxe manufacture existed in different populations, and b) that these were more similar between populations in geographic proximity than between those separated by substantial distances. This is likely to account for the emergent property of differences between large continental regions [22,88]. This is in line with previous findings that specific handaxe shapes may be representative of population history at specific places and times. For example, the British Acheulean record shows a temporal patterning of handaxe shape and reduction methods, likely reflecting the repeated source-sink dynamics of British occupation during the Pleistocene [46,48,50]. That morphological variability can further be patterned between different river valleys in England [47,50] hints at something corroborated by the findings of García-Medrano et al. [51]: that there is also a role for geographic connectivity between regions, rather than simple distance in structuring Acheulean populations. This has frequently highlighted for much later periods of time, such for the MSA of northern Africa [89], but it is now apparent this process has ancient roots.

We would argue that the cultural influences on final form are, therefore, most important across the wider archaeological record. Discussion of the cognitive underpinnings behind such cultural norms are beyond the scope of this paper (see e.g. [40,90–94]). Nonetheless, the established pattern of variation in handaxe shape between regions is perhaps not unexpected given evidence that shape plays no functional role in handaxe use. For example, Key and Lycett [95] have demonstrated that handaxe cutting efficiency is not related to known shape variation, and therefore shape may be free to drift to a much greater extent over large swathes of time and space.

## 4.2. Local drivers of handaxe variability

It is important to repeat that the results of the present study do not discount previous findings that suggest a role for blank type, raw material, or resharpening in handaxe morphology (e.g. [33,35–42,53,86,96–98]). We simply wish to stress that these differences are not regionally-patterned and therefore cannot explain the differences in morphology at broad continental scales (cf. [40]). This suggests that drivers of such variation are restricted to immediate knapping events, adding a degree of intra-assemblage variability (see Fig 18). These changes may also influence average differences between sites, but not across broad swathes of time and space. Indeed, the great inter-site variation in how refinement values change during the knapping sequence within each region may be consistent with previous research on much smaller spatial scales. For example, Shipton and Clarkson [43] report that the relationship between reduction intensity and relative thickness was not consistent between individual British sites in MIS13 and MIS11.

**4.2.1. Landscape habituation.** Aside from the influence of blank type, raw material, and resharpening, trends of landscape use may pattern the variability in plan symmetry and refinement seen within individual regions. This may include unique aspects of adaptation to local ecology, site function, or longer-term habituation to specific landscapes and accumulation of landscape knowledge (cf. [47,99–102]). With regards to landscape knowledge specifically, Clark and Linares-Matás [100–102] have argued that investment in lithic tool production is likely to have been greater as hominins inhabited individual landscapes for longer periods of time, allowing them to better understand the distribution of resources in time and space. In such circumstances, investment will be favoured because the returns are more predictable when occupying a novel or rapidly changing landscape. This may be reflected in the increasing refinement over time in the African sample, and the possible increases in plan symmetry, which feedback into longer term trends of cognitive evolution ([102]). This replicates the findings of other authors for specific landscapes, such as increased plan symmetry and refinement between Olduvai Bed II and Bed IV [53].

The lack of similar trends in Europe may be reflected in discontinuous source-sink occupation of northwestern Europe over time [103]—especially from MIS12 and the increasing intensity of glacial-interglacial cycles after the Mid-Bruhnes event (e.g. [96]). This would prevent very long-term accumulation of landscape knowledge, and limit the amount of investment in the stone tool assemblage. On the other hand, this period also sees a dramatic increase in the number of sites and in the diversity of materials and reduction schemes present in Europe, suggesting demographic changes associated with the increased intensity and duration of interglacials from MIS11 had a corresponding impact on hominin survival strategies (e.g. [87,97,104,105]). More consistent patterns of technological development may be expected for refugial areas of southern Europe that did not have to be abandoned during glacial periods, but these areas were undersampled in the present study.

With regards to the Levant, it is unclear if the landscape was occupied consistently across the Middle Pleistocene. Indeed, some authors have suggested that Levantine fauna tends to change alongside the greening and drying of the Sahara, resulting in occupation by African animals during wet periods, and European animals during dry ones (e.g. [106]). This may discourage consistent occupation of the Levant by one group of hominins, and instead reflect alternating dispersal from Africa and contraction from Europe. In such a scenario, the lack of trends in Levantine refinement and plan symmetry could be explained by a lack of long-term landscape habituation and knowledge accumulation. Nonetheless, these trends could equally be explained by continuous occupation, impacted by the nature of the later samples included in the present study. For example, the area surrounding Nadaouiyeh experienced desert

conditions across the Middle Pleistocene, as a result of an inability of the subsoil to retain water [76,79,107]. This would have been compounded by the extremity of harsh aridity during glacial periods, and a trend towards increasingly brackish conditions at the site's spring [108]. Under such ecological conditions, increased investment would not be expected over time due to a lack of predictability in resource distributions between years [102], although this does not rule out broader trends to be found elsewhere in the region over time (e.g. [109,110]).

**4.2.2. Site function.** Site function is particularly relevant to our potential finding that a continuum of handaxe reduction strategies existed between sites, but this must be discussed further as a novel hypothesis for explaining inter-assemblage variability at a local scale. Bowman's Lodge presented the greatest outlier for the relationship presented in Fig 17, as the only site for which handaxes clearly get more pointed as their tip length is reduced. Interestingly, Archer and Braun [111] showed that the intensity of reduction in South African handaxes was associated with PC1 scores suggestive of more pointed specimens, suggesting a potential alternative resharpening strategy for individual sites that may also have been followed at Bowman's Lodge. At the same time, Presnyakova et al. [112] suggest that this transition from more rounded to more pointed specimens represents the initial phases of knapping large flake blanks at Koobi Fora, before a shift towards subsequent rounding of the handaxe outline towards the end of the knapping sequence. As such, the relationship between the resharpening coefficient and Tip Length at Bowman's Lodge may equally be explained by a bias towards discard of handaxes in early-to-intermediate phases of knapping at the site. In either case, Bowman's Lodge appears to be rather atypical for the Acheulean assemblages included in the present study, and does not seem to be the most illustrative example of the wider relationship between sites.

Leaving Bowman's Lodge to one side, a possible counter to the wider trend of Fig 17 is that the sites on its far left are simply earlier phases of reduction, and that resharpening has not acted on these specimens yet. This would require a relationship in which the longest of the initial blanks were initially the thinnest relative to their width, before resharpening acted to thin the artefacts at a rate greater than reduction of the length. In contrast to such a claim, a sample of 60 cobbles and 60 large flakes—all on flint—produced for an unrelated experiment [82] shows no such relationship between length and refinement on either blank type (Cobbles: r = -.038, $p$ = 0.763; Flakes: r = -.044, $p$ = 0.7404). This suggests the negative association between length and refinement values (suggesting longer handaxes are thinner relative to their width compared to shorter ones) on the left side of Fig 17 is a feature imposed by reduction, rather than a starting point.

The driving force behind also maintaining tip shape for these specimens can be explained in at least two ways: that the knapper was trying to maintain the "forward extension" [cf. 30,113] of artefacts to maintain loading during cutting tasks [114], or that they were trying to maintain an elongated platform for flake release [115]. Experimental work does suggest that increasing edge angles (likely associated with increased thickness relative to width [53]) up to ~70˚ is not detrimental to handaxe cutting performance, and may actually improve it [81,116]. This may suggest that knappers can choose to retain the shape of artefacts as they get relatively thicker, to maximise the force that could be loaded onto the tip [117]. Given the pattern in which the negative length-refinement relationship is strongest in magnitude when the resharpening coefficients in Fig 17 are weakest, such a functional consideration would imply a trade-off in handaxe manufacture, either making the artefact relatively thicker to increase the force that can be exerted at the tip, or changing the artefact shape to maximise the amount of available cutting edge [38,39]. At the same time, it is not obvious as to why these two strategies would be mutually exclusive, given reducing the width quicker than the thickness is not an obstacle to changing tip shape during reduction, and changes to tip shape are not associated with differences to the force applied or cutting efficiency [95,117].

A greater focus on the release of flakes towards the left of Fig 17 may provide an alternative explanation for the association between length-refinement coefficients and changes to tip shape across knapping. We may be able to test this through examination of archaeological signatures of site function, and how these map onto site-level placement along the continuum. For example, both Olorgesailie Member 1 and Boxgrove show very similar discard patterns within the landscape, in which discard is greatest near raw material sources and pathways out of the exploited landscape [14,118]. This may suggest commonalities in its use despite dramatic differences in palaeoenvironment. Furthermore, given their central position on the scatterplot, our model would predict that both flake production and the use of a cutting edge were important goals of these assemblages. Indeed, both sites show clear evidence for use of flakes produced from handaxe manufacture. At Olorgesailie, Potts [14,119,120] has concluded that handaxes were produced near to raw material sources and subsequently carried into the lake basin to be used as a source of flakes in animal exploitation. Of particular interest is Site 15 on the lake margin palaeosols, in which an *Palaeooxodon* (formerly *Elephas*) *reckii* skeleton was found in direct association with hundreds of flakes whose flake scars clearly attest to being struck from handaxes, but very few handaxes themselves [119,120]. Handaxes were thus carried away from the site, and appear to have frequently been discarded in fluvial channel contexts, which Potts et al. [14] suggest provided pathways out of the lake basin when rivers dried up. With regards to Boxgrove, cores are extremely rare, with none producing flakes that could not also result from handaxe reduction [121]. High resolution confirmation for the use of handaxes flakes can be provided by the sites in Unit 4b, where rapid low-energy deposition preserves isolated activity episodes. At Q1/A, the débitage suggests a single episode of handaxe reconfiguration, with the largest flakes selectively moved to one side by the knapper [121]. At GTP-17, flakes from handaxe manufacture were carried around the site and show evidence of direct use during a single horse (*Equus ferus*) butchery event [121,122]. Unfortunately, evidence for (or against) direct use of the Olorgesailie handaxes is currently limited, but use of Boxgrove specimens is strongly supported by use-wear evidence [123,124]. Indeed, the frequent tranchet blows to the Boxgrove handaxes may be an adaptation to maximise torque at the tip of the handaxe [117].

Notarchirico is also illustrative because handaxes at the site may have been deposited in the context of possible individual activity episodes (although there is no clear evidence of anthropogenic interaction with bone), with artefacts manufactured from local cobbles retrieved from an immediately local lakeshore [125–129]. In such a scenario, it would be unlikely that the artefacts were transported far into the site, and that there was instead a rather ad-hoc production of bifaces for use as cutting tools (i.e. short-term occupations). In turn, we would expect that discarded handaxes would have only seen a short life history, and thus the likelihood of resharpening would be low. Indeed, this is consistent with the very limited technological evidence for resharpening at the site [126], as well as the limited evidence for changes to tip shape in the present study. It has been speculated that at least one of the levels at Notarchirico for which data were included here (layer F) might reflect a longer-term domestic occupation [129], but the accumulation may equally relate to recurrent but individual episodes of behaviour [128]. In this context, our findings are in line with expectations for the site, and also suggest that distinct pressures are driving the relationships between length and refinement on one hand, and resharpening on the other. As such, site-specific pressures may account for specific deviations from the regression line of Fig 17.

The existence of continuum of handaxe reduction strategies would suggest handaxes represent a generalist tool that facilitated flexible deployment in a wide range of circumstances, particularly in the face of habitat and climatic variation. In terms of the latter, such flexibility may be an adaptation to multi- (e.g. [120,130–133]) and/or single-generation inter-annual and

seasonal (e.g. [17,102,134–139]) pressures. This may give one possible reason for its persistence over such long swathes of time, if more specialised tools offered little advantage in particular tasks or situations.

**4.2.3. Handaxes and dispersal.** Sites from each region are also distributed across Fig 17, which would imply any continuum of strategies was present in handaxe-producing populations of East Africa before any dispersal of the technology out of the continent. However, if this is the case, it raises the obvious question as to why handaxes did not become established outside Africa sooner. North of Africa, the presence of handaxes before 1 Ma is largely limited to 'Ubeidiya in Israel [140], and they do not become ubiquitous in the Levant and Europe until after 0.8 Ma (with sites such as GBY in the Levant and Notarchirico, La Noira, and Moulin Quignon in Europe [141–147]. One such possibility is that the adaptive flexibility witnessed in reduction strategies was not present from the beginning of the Acheulean, and was only innovated later in the Early Pleistocene. Indeed, Moncel et al. [32] have used the data from Moulin Quignon to suggest that behavioural flexibility may have been key in facilitating dispersal into the extreme cold (glacial) environments of northern Europe at the beginning of the Middle Pleistocene. This may represent a difference in adaptability compared to hominins in northern Europe during the Early Pleistocene. In such a scenario, early dispersing hominins may have relied upon the flexibility offered by core-and-flake technologies in landscapes for which they had limited knowledge [100,115], whereas later Acheulean dispersals may have relied on technologies that perhaps reduced the knowledge barrier for entrance into novel environments (see [148]). Such a question will only be approached by an increased sampling of specimens from the Early Acheulean of East Africa, as a likely source of dispersal.

## 5. Conclusions

The data we present here strongly suggest that average handaxe shape varies between the continental regions of East Africa, the Levant, and Europe, and that these differences are likely to be largely independent of blank type, raw material use, and the extent of resharpening. Instead, it seems likely that regional patterning emerges largely as the result of local norms of handaxe manufacture, and that these were, on average, more similar between neighbouring groups than more distant ones. This may relate to a number of mechanisms, including cultural diffusion of manufacture norms and connectivity of individual regions, and/or greater temporal proximity to manufacture norms in a shared ancestral population.

In contrast, plan symmetry and resharpening do not clearly vary at the same spatial scales, likely showing a greater influence of local ecology, site function, and/or long-term landscape habituation. It may be argued that the functional implications of handaxe refinement prevented it from experiencing drift in the same way as overall shape. In particular, the specific reduction strategies employed in a given assemblage is unlikely to have been random, with hominin populations able to produce a range of outcomes related to the function of the tool. This may have ranged from a focus on the maintenance of tip shape and the production of flakes, to a focus on the production and maintenance of a sharp cutting edge, and have facilitated substantial behavioural flexibility in the deployment of bifaces across time and space. Further experimental and archaeological data is needed to validate the assumptions of this model, particularly in relation to core-reduction strategies.

## Supporting information

**S1 File. Supplementary data, methodology, and site descriptions.** S1 Table. Supplementary Data from the EFA PCA, additional methodological data, and descriptions of the sites included

in the analysis.
(PDF)

**S2 File. Artefact coordinate data.** 2D coordinate data for all of the artefacts included in the analysis.
(XLSX)

**S3 File. Artefact attributes and PCA data.** All remaining data used in the analysis, including categorical, morphometric, and principal component data.
(XLSX)

## Acknowledgments

The work submitted in this paper was carried out as part of the MPhil project of JC, whose work was supported by the Cambridge Trust (through a UK Master's Scholarship), the University of Cambridge Department of Archaeology (through a departmental Studentship), and by St. John's College, University of Cambridge (through a Graduate Scholarship). We are extremely grateful to the various insitutions who allowed access to materials for the completion of this work, including the Musée de l'Homme and Institut de Paléontologie Humaine of the Muséum National d'Histoire Naturelle (Paris, France), the Museo archeologico nazionale di Venosa (Venosa, Italy), the Museum of Archaeology and Anthropology (Cambridge, UK), the British Museum (London, UK), and the University of Basel unit of Integrative Prehistory and Archeological Science (Basel, Switzerland). We extend a special thanks to Reto Jagher, who facilitated access to the Nadaouiyeh Aïn Askar material, and to Gadi Herzlinger and Naama Goren-Inbar, who generated the images of the Gesher Benot Ya'aqov handaxes. We also thank the National Museums of Kenya for access to the artefacts from Olorgesailie and Isinya, photographed as part of a previous project by CS.

## Author Contributions

**Conceptualization:** James Clark, Philip Ronald Nigst, Robert Andrew Foley.

**Data curation:** James Clark.

**Formal analysis:** James Clark.

**Funding acquisition:** James Clark.

**Investigation:** James Clark, Ceri Shipton.

**Methodology:** James Clark.

**Project administration:** James Clark.

**Resources:** Marie-Hélène Moncel.

**Supervision:** Philip Ronald Nigst, Robert Andrew Foley.

**Visualization:** James Clark.

**Writing – original draft:** James Clark.

**Writing – review & editing:** Ceri Shipton, Marie-Hélène Moncel, Philip Ronald Nigst, Robert Andrew Foley.

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
