## [Decision Letter · Decision Letter 0]

21 Feb 2024

PONE-D-23-29028When is a handaxe a planned-axe? Exploring morphological variability in the AcheuleanPLOS ONE

Dear Dr. Clark,

Thank you for submitting your manuscript to PLOS ONE. After careful consideration, we feel that it has merit but does not fully meet PLOS ONE’s publication criteria as it currently stands. Therefore, we invite you to submit a revised version of the manuscript that addresses the points raised during the review process.

We look forward to receiving your revised manuscript.

Kind regards,

Enza Elena Spinapolice, Ph.D

Academic Editor

PLOS ONE

Journal Requirements:

3. Please include a complete copy of PLOS’ questionnaire on inclusivity in global research in your revised manuscript. Our policy for research in this area aims to improve transparency in the reporting of research performed outside of researchers’ own country or community. The policy applies to researchers who have travelled to a different country to conduct research, research with Indigenous populations or their lands, and research on cultural artefacts. The questionnaire can also be requested at the journal’s discretion for any other submissions, even if these conditions are not met.  Please find more information on the policy and a link to download a blank copy of the questionnaire here: https://journals.plos.org/plosone/s/best-practices-in-research-reporting. Please upload a completed version of your questionnaire as Supporting Information when you resubmit your manuscript.

4. We note that you have referenced (ie. Bewick et al. [5]) which has currently not yet been accepted for publication. Please remove this from your References and amend this to state in the body of your manuscript: (ie “Bewick et al. [Unpublished]”) as detailed online in our guide for authors

Reviewers' comments:

Reviewer's Responses to Questions

**Comments to the Author**

1. Is the manuscript technically sound, and do the data support the conclusions?

Reviewer #1: Yes

Reviewer #2: Partly

2. Has the statistical analysis been performed appropriately and rigorously? 

Reviewer #1: Yes

Reviewer #2: Yes

3. Have the authors made all data underlying the findings in their manuscript fully available?

Reviewer #1: Yes

Reviewer #2: Yes

4. Is the manuscript presented in an intelligible fashion and written in standard English?

Reviewer #1: Yes

Reviewer #2: Yes

5. Review Comments to the Author

Reviewer #1: I think this paper is really interesting and should be considered to be published within this journal. The topic is very attractive, as well as the sites included for the analysis. The data support the conclusions, the materials are appropriated, as well as the methods used. Statistically is well supported and the manuscript is well-written in a correct English. I think all the tables and figures and also adequate for the complete understanding of the paper, and even the Supplementary Materials. By all these reasons, I strongly recommend its publication with major revisions.

Materials and Methods

2.1 Sample: Authors said they have collected data from plan a d profile view, but then they only use plan-shape data to make the geometric morphometrics study. Profile views are only used from a metrical point of view. It is something key within this paper. Without any doubt, paper will be better justified and clearly written if profile-shape was explored.

Table 1. Other significant problem is no to include European sites from MIS11 where authors could test the coarse-grained materials: e.g. Menez Dregan, Terra Amata or other Spanish sites.

There is not technological analysis at all, explained in the Methods section. From my point of view, to make an accurate morphometrical analysis to have characterized technologically the sample. You should quantify the technological data to describe the materials. At least, it is necessary to include those aspects considered within this paper. What is resharpening? How has it been measured? What is thinning? How has it been measured?

There are no images, at all. You cannot check what the authors are explaining and this fact, combined with no technological analysis, made almost impossible to figure out what authors are referring to.

In this sense, it would be very helpful if Figure 3 could be integrated in the Figure 4, to visualize the morphological tendencies of each PCs.

Results

It is said that PC1 “represents elongation and tip shape. Higher scores reflect more elongated and pointed handaxes, and lower scores less elongated and more ovate forms”. Nevertheless, this PC represents also the variation in the whole width: distal width, mid width and proximal width. The Excel contains only one “width”, I assume the maximum width. It could be interesting to include the other widths. If you do this, you can calculate features such as the refinement of the tip (B1/T1), and reinforce or qualify your conclusions.

Here is where I would need to see the profile-shape analysis. I recommend strongly to introduce it. Using 2D images it is mandatory to make plan-shape and profile-shape analysis independently. But I am sure this paper will benefit a lot if authors accept to include this analysis.

3.1. Spatial variation. This title is quite intriguing. I suggest authors to change it by “variation in shape”/”shape variation” or something like this.

3.1.1. Shape = Plan-shape. It should be specified.

In the first line (265) symmetry is plan-symmetry. It should be specified along the whole test. Symmetry is traditionally associated to profile. As this is not the case, please clarify it.

Line 274-275: Refinement can also be significantly distinguished between regions (F = 12.467, p = 4.969e-275 06), driven by relatively thicker handaxes in the African sample. We cannot see it very well. I’d need the mean thickness by “area” and Supplementary Information has the Excel in a tool-by-tool list. It can also make reference to Figure 10. But if so, this Figure is reflecting major refinement in a great part of the African register (as recorded in Line 394). It seems to be contradictory.

Line 302 to 311. This is an interpretation of the data, and in a very personalized way. Perhaps all this paragraph should be moved to Discussion. But in any case, European sites are huge assemblages, too. Try to distinguish which sample is more palimpsest that the other is a mistake. Perhaps, it is better to consider that many African sites has a minimum age and most of the European sites have relative datings or big error ranges in absolute datings.

3.2.2 Symmetry = Plan-symmetry

3.2 Blank Type

It is necessary to include a Table with all information about the number of tools by type of tool, type of blank, etc.

Authors talk about cobble/flake. But what about the indeterminate blanks? A table would be so useful.

“Cobbles are overrepresented in Europe and Levant and are the thicker instruments”, so they introduce variability. But if you exclude cobbles to analyse flakes, it is something highly probable to find major uniformity, no? I think this subsection is a little bit redundant.

Fig 10. From Africa to Levant and Europe: those areas are chronologically complementary, mainly from MIS12, when all changes happens. In Europe there is an increase of diversity, major technological flexibility, also with the massive use of coarse-grained materials, together with a higher regionalization of technology. African register cannot show it because there are no included younger sites. On the contrary, if authors do not consider European sites with coarse-grained materials, you are avoiding a significant part of the register. As authors stated in lines 454-455, “significant differences emerged driven by thicker artefact made on quartzite and lava”.

Apart from Figure 11, it is necessary to see the distribution by zones but including cobble information.

3.4 Please, define Resharpening in the method section

3.4.2 Thinning: here is where a deep shape analysis of the profiles could be crucial. Because there is a clear association between thinning works and modification of the edge angle (García-Medrano et al. 2023).

Besides, from my point of view, the use of “cores” (choppers and discoidal cores) to compare with handaxes is not appropriated to get conclusions about resharpening and refinement. Basically, because resharpening is not the same than recurrence in flaking. This is the reason because the conclusions of this section are so obvious. Literally, authors said “Unsurprisingly”. (Line 567)

I think there is a mistake in the reference 42 (Line 565). It should be 44 or 45.

Regarding Figure 16, there are well-known classical sites in contradictory positions to what authors conclude form this Figure: continuum of possible site-level reduction strategies, ranging from one more focused on the maintenance of tip shape and the production of flakes on the left side of the distribution, to one more focused on the production and maintenance of a sharp cutting edge on the right side of the distribution.

Boxgrove or Elveden appear on the left and are the classical oval and tear-drop shape. On the contrary, warren Hill appears totally on the right, and it is well know that has both oval and pointed handaxes.

Besides the r value is not so high 0.302 to justify any strong correlation.

Please, consider to review in deep this site distribution.

Discussion

I think this section should carefully be reviewed by authors. It is necessary to order the ideas to be discussed and use the data presented in the result section to justify the conclusions. It is complicated to follow the structure and there are a lot of crucial aspects lost mainly in subsection 4.1.

Finding 5 in Table 3. According to Figure 10, there is a decrease in the African refinement, contrary to what it is stated here. The same in Line 687. Please review carefully this.

Reviewer #2: The paper submitted by J. Clark et al. propose a comparison, based on a morphometric measurement, of more than 1.000 Lower Palaeolithic handaxes of Africa, Levant, and Europe. The authors choose a very specific method, that to be perfectly honest I'm not very familiar with. They have also chosen to cite only a very selective bibliographical reference, overwhelmingly from a specifical epistemological tradition. For example, there is almost no reference to lithic technology or use wear analysis. And they have every right to do so! But this makes the review rather complicated, especially the method section. Having said that, I will try to do a constructive review by questioning the compliance of the paper with scientific standards.

1. Introduction

The introduction is quite efficient, even if the definition of Acheulean (Acheulean = handaxes) is a bit of a caricature and could have been a little more nuanced (An extremely hot topic of debate!).

Then, the authors set out the various hypotheses that could explain the differences between handaxes but leave out the most obvious possibility: that they were different tools.

The research questions are exposed in the last paragraph “This may better allow us to understand the reasons for handaxe persistence over time, and therefore how they related to the survival strategies of the hominins that produced and used them.” This is in my opinion the biggest problem with the paper: how can this approach answer this question? How can the comparison between the shapes of the handaxes explain why they have lasted so long? Even though I'm repeating it, I'm not familiar with the method, this point needs to be clarified. How can the comparison of handaxes shapes contribute to the Acheulean issue?

2. Materials and Methods

The presentation of the archaeological sample is clear, well exposed in the Table 1. However, the question of the choices made is never addressed. Why these pieces and not others? Both chronologically and geographically.

Concerning the measurement strategy, I don't really have any comments, except that I'm wondering why the sections aren't considered, even though sections are crucial in the handaxe structure.

3. Results

The presentation of the raw results is clear and didactic. Unfortunately, when they try to interpret the results, things get complicated. The authors try to make correlations between morphometric data and thinning, refinement and resharpening which seems rather unexpected, and moreover on the basis of a few pieces from completely different contexts.

4. Discussion

The discussion is highly developed and puts the results into perspective. However, we move on from extremely general and partial results (some measurements on some piece) to micro-local considerations, difficult to reach in the Lower Palaeolithic context (environment, site function, subsistence activity, mobility, etc). This radical gap between this tzo scales seems difficult to reconcile and with contrary to the initial ambition of the paper.

5. Conclusion

Unfortunately, the results are disappointing: « Instead, it seems likely that regional patterning emerges largely as the result of local norms of handaxe manufacture, and that these were, on average, more similar between neighbouring groups than more distant ones ». Moreover, this result was expected given the lack of chronological and geographical diversity within each region.

To sum up, I would say that the work carried out is important and that it would be a shame not to publish the results. To be published, we suggest to

- discuss the relevance of the selected pieces: why we are comparing this handaxe with this other one?

- adapted the research question to the method: what results can we expect when we compare tool shapes?

6. PLOS authors have the option to publish the peer review history of their article (what does this mean?). If published, this will include your full peer review and any attached files.

Reviewer #1: No

Reviewer #2: **Yes: **Roxane Rocca

---

## [Author Response · Author response to Decision Letter 0]

29 Apr 2024

Dear Dr. Spinapolice, Dr. Rocca, and one anonymous reviewer,

Thank you all so much for taking the time to review our paper, and for your kind words about its content. We really appreciate the detail with which you each commented on the paper, and are confident it has become a better piece of research because of it.

The reviewers broadly agree that the paper has value in its contribution to the literature, that the statistical analyses have been performed appropriately and rigorously, that the data have been made fully available, and that the manuscript is intelligible and written in standard English, but provide specific suggestions regarding its framing, methodology, and presentation of the results. We think these are cogent suggestions for the publication of the research, most of which we have tried to integrate in full. At the same time, we wish to comment in more detail on a couple of the points raised by the reviewers before we respond to the reviews more thoroughly.

Reviewer 1 (and briefly, also, Reviewer 2) correctly points out that the paper focuses on plan shape, and recommends an analysis of the profile photos using the same methodology to compliment the research. We completely agree that the side-view photos would be very informative in the context of our research, especially in providing finer-grained information on bilateral symmetry and the nature of thinning across the length of the piece. This work is planned for future work on the dataset, but we chose not to include it in this paper for a few main reasons, that form the basis of our wish to keep such data outside of the current manuscript.

First of all, while we acknowledge the reduced resolution necessary when using only plan-form in handaxe analyses (e.g. Iovita et al., 2017; Key, 2019; García-Medrano et al., 2020), a focus on 2D plan-view images has an established tradition in Acheulean geometric morphometric analysis (e.g. Iovita, 2009; Costa, 2010; Iovita and McPherron, 2011; McNabb, 2017; Hoggard et al., 2019; Key et al., 2023). These studies have demonstrated the value of studying 2D plan-shape in isolation from its profile counterpart, and we believe it has real utility in the current paper. Secondly, not all of the sites contain profile images, and all of those are from the African sample (i.e. any site from Olorgsailie, Isinya, and Kharga Oasis). This is already the smallest subsample in the analysis, and therefore taking out this series of would restrict the broader utility of any inter-regional analysis looking at side-profile. We believe this would unnecessarily detract from the differences in plan-view shape identified in this paper. Finally, the image processing stage (going from the images to measurements and silhouettes to coordinates) is particularly intensive, and would delay publication of the research by at least a year. We therefore hope the reviewers understand our decision not to include such an analysis at this stage.

In a similar vein, both Reviewer 1 and Reviewer 2 also highlight the lack of technological analysis of artefacts included in the paper, and we completely understand the archaeological traditions that underlie this recommendation, as well as the methodological advantages that come from combining such analyses with geometric morphometric studies of Acheulean tool form (García-Medrano et al., 2021). Nonetheless, we wish to highlight that that that the paper is derived from an established set of methods that aim to look at shape independent of finer-grained technological analyses, to approach much broader macro-evolutionary questions than can be answered from individual artefacts themselves. This was the aim from the very beginning of the research, which therefore meant that data collection prioritised the number of artefacts photographed, and no formal typological or technological research was carried out. This resulted in the very large sample size compiled in the paper, but inevitably presents a tradeoff between the depth of information gleaned from individual artefacts, and what can be established from the sample as a whole. At the same time, much of the technological information already published for sites in the analysis is reiterated as context in the Supplementary Information.

Reviewer 1 further underlines the fact that a number of technological terms have not been adequately defined (including “resharpening” and “thinning”), and we have done our best to rectify this shortfall. Nonetheless, resharpening in particular has been repeatedly hypothesised to reflect changes to 2D shape over the knapping sequence (e.g. McPherron, 1995, 1999, 2000, 2003, 2006; Iovita and McPherron, 2011), and remains an important topic of discussion (e.g. Shipton and Clarkson, 2015a, 2015b; Iovita et al., 2017; García-Medrano et al., 2019; Shipton and White, 2020). Given the issues highlighted in the paper with translating the resharpening relationship hypothesised by McPherron into geometric morphometric analyses, we believe our paper highlights the continued utility of looking at artefacts from the assemblage-scale and beyond, without the need to always collect more labour-intensive data in the form of 3D scans or manual analysis of the tool (e.g. Scar Density Indices). This may provide a basis for a more formal examination of this relationship and technological data from individual artefacts in the future.

We wish to reiterate our appreciation for the comments provided by the reviewers, and their willingness to engage with our research. We believe that these comments have made our research better, and hopefully make our manuscript better suited for publication. We present a point-by-point response to each reviewer below, with our responses and adjustments in black. 

We look forward to hearing back from each of you.

Kind regards,

James Clark

Point-by-Point Response

Reviewer 1

I think this paper is really interesting and should be considered to be published within this journal. The topic is very attractive, as well as the sites included for the analysis. The data support the conclusions, the materials are appropriated, as well as the methods used. Statistically is well supported and the manuscript is well-written in a correct English. I think all the tables and figures and also adequate for the complete understanding of the paper, and even the Supplementary Materials. By all these reasons, I strongly recommend its publication with major revisions.

We thank the reviewer for their kind words and recommendation of our paper for acceptance, pending our address of their comments.

Materials and Methods

2.1 Sample: Authors said they have collected data from plan a d profile view, but then they only use plan-shape data to make the geometric morphometrics study. Profile views are only used from a metrical point of view. It is something key within this paper. Without any doubt, paper will be better justified and clearly written if profile-shape was explored.

Our response to this is laid out in more detail above, but we reiterate our agreement that such an analysis will provide useful complimentary information in futureEe have also added the following important caveat surrounding the exclusion of the profile images in this way:

“Geometric Morphometric Analysis was carried out on all plan-view images, with the workflow of preparation for this is summarised in Figure 2. We are aware that not applying the analysis to profile images removes power from the analysis, especially when discriminating between tool forms (e.g. Iovita et al., 2017; Key, 2019; García-Medrano et al., 2019), but follow an established tradition of 2D biface shape analysis (e.g. Iovita, 2009; Costa, 2010; Iovita and McPherron, 2011; McNabb, 2017; Hoggard et al., 2019; Key et al., 2023).”

Table 1. Other significant problem is no to include European sites from MIS11 where authors could test the coarse-grained materials: e.g. Menez Dregan, Terra Amata or other Spanish sites.

We completely agree that there is a clear sampling bias in the European data towards the absence of sites with more coarse-grained material, especially those that might relate to the “Large Flake Acheulian” of Sharon (2007, 2009, 2010). This is ultimately a reflection of ease of access to samples (either in museums to be accessed in early 2020 or already-published information following the COVID-19 lockdown) across time and space, rather than an intentional decision by us. As mentioned below, there is also an underrepresentation of eastern African samples from the Middle Pleistocene. We have added a caveat to the presentation of the sample description in section 2.1 to underline this difficulties, as well as the following to the caption of Table 1:

“There is an unfortunate dearth of assemblages on coarse-grained materials from Europe, especially from the Iberian peninsula, as well as of eastern African assemblages from the Middle Pleistocene.”

There is not technological analysis at all, explained in the Methods section. From my point of view, to make an accurate morphometrical analysis to have characterized technologically the sample. You should quantify the technological data to describe the materials. At least, it is necessary to include those aspects considered within this paper. What is resharpening? How has it been measured? What is thinning? How has it been measured?

Our detailed response as to why no new technological analysis is carried out in this paper is outlined above, but it is worth noting that a substantial amount of technological information is summarised for a number of the assemblages in the Supplementary Material, to help characterise the assemblages in the way suggested by the Reviewer. This information is included as part of the site descriptions in section 3. In order to make this more explicit, we have modified lines 166-167 to read as:

“For more detailed descriptions of site stratigraphy, dating, sample composition, and technological data, see the Supplementary Information.”

With regards to resharpening, specifically, we again reiterate that we were focusing on the allometric relationship attributed to resharpening by McPherron (1995, 1999, 2000, 2003, 2006; Iovita and McPherron, 2011). We have therefore made sure to include a section (2.4) entitled “Allometry” in the the methods (that now corresponds to a section with the same name in the results), which outlines the relevant background literature and the relationships being explored. We have also changed the headings of 3.4.1 and 3.4.2 to “Changes to Plan-Shape” [previously “Shape”] and “Changes to Thickness” [previously “Thinning”], respectively, to make it clearer as to which phenomena we are referring. And finally, we have reworked the presentation of the results in section 3.4 to better reflect the logical flow of the manuscript and make our points clearer.

There are no images, at all. You cannot check what the authors are explaining and this fact, combined with no technological analysis, made almost impossible to figure out what authors are referring to.

In this sense, it would be very helpful if Figure 3 could be integrated in the Figure 4, to visualize the morphological tendencies of each PCs.

We understand the difficulty with visualisation at points throughout the manuscript, and have now modified Figure 4 to integrate the PC changes from Figure 3. We have not felt able to the same with other figures, because it is more difficult to integrate when PC values are being displayed on a single axis (especially when that axis is displaying multiple PCs). This is why we felt it important to create Figure 3 as a standalone plot that can be referred back to throughout.

Results

It is said that PC1 “represents elongation and tip shape. Higher scores reflect more elongated and pointed handaxes, and lower scores less elongated and more ovate forms”. Nevertheless, this PC represents also the variation in the whole width: distal width, mid width and proximal width. The Excel contains only one “width”, I assume the maximum width. It could be interesting to include the other widths. If you do this, you can calculate features such as the refinement of the tip (B1/T1), and reinforce or qualify your conclusions.

Here is where I would need to see the profile-shape analysis. I recommend strongly to introduce it. Using 2D images it is mandatory to make plan-shape and profile-shape analysis independently. But I am sure this paper will benefit a lot if authors accept to include this analysis.

Again, our response here is outlined in more detail above, but it is worth touching on a couple of points. We completely agree that these data would be useful alongside an analysis of the profile images, and therefore may make a particularly useful addition to a future paper on this. At the same time, while it is true that PC1 also reflects changes to the distribution of material at the different widths, we do not feel these measurements would add much to the paper in its current form. Indeed, elongation and tip shape together account for 94% of the variance in PC1 in the subset of handaxes for which independent metrics of each are available. This indicates only a very small proportion of additional variance would be explained by having the other metrics of width and thickness, and would not add information for the former that cannot be visualised in the PCA. 

3.1. Spatial variation. This title is quite intriguing. I suggest authors to change it by “variation in shape”/”shape variation” or something like this.

We have changed the title to “Regional Variation” to acknowledge that the spatial differences we are talking about are rather coarse-grained, but have chosen not to include “shape” in this part of the title because the section includes both plan shape and symmetry.

3.1.1. Shape = Plan-shape. It should be specified.

Agreed and changed.

In the first line (265) symmetry is plan-symmetry. It should be specified along the whole test. Symmetry is traditionally associated to profile. As this is not the case, please clarify it.

We have changed almost all references to symmetry in the paper to specify we are referring to plan symmetry.

Line 274-275: Refinement can also be significantly distinguished between regions (F = 12.467, p = 4.969e-275 06), driven by relatively thicker handaxes in the African sample. We cannot see it very well. I’d need the mean thickness by “area” and Supplementary Information has the Excel in a tool-by-tool list. It can also make reference to Figure 10. But if so, this Figure is reflecting major refinement in a great part of the African register (as recorded in Line 394). It seems to be contradictory.

I think the reason the trend cannot be seen so well in Figure 10 is because of the potential trend described for the African sample in which sites become more refined over time. However, those sites also have lower sample sizes, so there is ultimately a skew towards the earlier sites when comparing between regions. This information is not contradictory between sections. We have added the following to make this clearer in the text:

“Refinement can also be significantly distinguished between regions (F = 12.467, p = 4.969e-06), driven by relatively thicker handaxes in the African sample, likely related to the much smaller sample sizes in the later African assemblages (see section 3.2.).”

In addition, plan-form area data for each of the artefacts were not collected, largely because Roe’s “refinement” values are well established in the field, using width as the main axis of size through which thickness can be standardised (e.g. Shipton, 2018). The different sizes of the input images before Procrustes superimposition also means that centroid size can also not be used as an unbiased marker of size. This measurement of refinement is present for each tool in column L of the excel spreadsheet.

Line 302 to 311. This is an interpretation of the data, and in a very personalized way. Perhaps all this paragraph should be moved to Discussion. But in any case, European sites are huge assemblages, too. Try to distinguish which sample is more palimpsest that the other is a mistake. Perhaps, it is better to consider that many African sites has a minimum age and most of the European sit

---

## [Decision Letter · Decision Letter 1]

1 Jul 2024

When is a handaxe a planned-axe? Exploring morphological variability in the Acheulean

PONE-D-23-29028R1

Dear Dr. Clark,

We’re pleased to inform you that your manuscript has been judged scientifically suitable for publication and will be formally accepted for publication once it meets all outstanding technical requirements.

Kind regards,

Enza Elena Spinapolice, Ph.D

Academic Editor

PLOS ONE

Additional Editor Comments (optional):

Reviewers' comments:

Reviewer's Responses to Questions

**Comments to the Author**

1. If the authors have adequately addressed your comments raised in a previous round of review and you feel that this manuscript is now acceptable for publication, you may indicate that here to bypass the “Comments to the Author” section, enter your conflict of interest statement in the “Confidential to Editor” section, and submit your "Accept" recommendation.

Reviewer #1: All comments have been addressed

Reviewer #2: All comments have been addressed

2. Is the manuscript technically sound, and do the data support the conclusions?

Reviewer #1: Yes

Reviewer #2: Yes

3. Has the statistical analysis been performed appropriately and rigorously? 

Reviewer #1: Yes

Reviewer #2: Yes

4. Have the authors made all data underlying the findings in their manuscript fully available?

Reviewer #1: Yes

Reviewer #2: Yes

5. Is the manuscript presented in an intelligible fashion and written in standard English?

Reviewer #1: Yes

Reviewer #2: Yes

6. Review Comments to the Author

Reviewer #1: (No Response)

Reviewer #2: I thank the authors for the attention paid to our remarks and for the precise answer. I remain very doubtful about the interest of this methodological approach for providing knowledge on Lower Palaeolithic societies. But I recognize that the authors took each comment into account and responded to the criticisms in a convincing manner. I am therefore agree with the publication of this article in the journal.

7. PLOS authors have the option to publish the peer review history of their article (what does this mean?). If published, this will include your full peer review and any attached files.

Reviewer #1: No

Reviewer #2: No

---

## [Editor Report · Acceptance letter]

5 Jul 2024

PONE-D-23-29028R1 

PLOS ONE

Dear Dr. Clark, 

I'm pleased to inform you that your manuscript has been deemed suitable for publication in PLOS ONE. Congratulations! Your manuscript is now being handed over to our production team.

Kind regards, 

on behalf of

Dr. Enza Elena Spinapolice 

Academic Editor

PLOS ONE